

# Overview of the PALM model system 6.0

Björn Maronga[1,2], Sabine Banzhaf[3], Cornelia Burmeister[4], Thomas Esch[5], Renate Forkel[6], Dominik Fröhlich[7], Vladimir Fuka[8], Katrin Frieda Gehrke[1], Jan Geletič[9], Sebastian Giersch[1], Tobias Gronemeier[1], Günter Groß[1], Wieke Heldens[5], Antti Hellsten[10], Fabian Hoffmann[1,11], Atsushi Inagaki[12], Eckhard Kadasch[13], Farah Kanani-Sühring[1], Klaus Ketelsen[14], Basit Ali Khan[6], Christoph Knigge[1,13], Helge Knoop[1], Pavel Krč[9], Mona Kurppa[15], Halim Maamari[16], Andreas Matzarakis[7], Matthias Mauder[6], Matthias Pallasch[16], Dirk Pavlik[4], Jens Pfafferott[17], Jaroslav Resler[9], Sascha Rissmann[17], Emmanuele Russo[3,18,19], Mohamed Salim[20,21], Michael Schrempf[1], Johannes Schwenkel[1], Gunther Seckmeyer[1], Sebastian Schubert[20], Matthias Sühring[1], Robert von Tils[1,4], Lukas Vollmer[22,23], Simon Ward[1], Björn Witha[22], Hauke Wurps[22], Julian Zeidler[5], and Siegfried Raasch[1]

[1]Leibniz University Hannover, Institute of Meteorology and Climatology, Hannover, Germany
[2]University of Bergen, Geophysical Institute, Bergen, Norway
[3]Freie Universität Berlin, Institut für Meteorologie, Berlin, Germany
[4]GEO-NET Environmental Services GmbH, Hannover/Dresden, Germany
[5]German Aerospace Center (DLR), German Remote Sensing Data Center (DFD), Oberpfaffenhofen, Germany
[6]Karlsruhe Institute of Technology, IMK-IFU, Garmisch-Partenkirchen, Germany
[7]Deutscher Wetterdienst, Research Center Human Biometeorology, Freiburg, Germany
[8]Charles University, Department of Atmospheric Physics, Faculty of Mathematics and Physics, Prague, Czech Republic
[9]The Czech Academy of Sciences, Institute of Computer Science, Prague, Czech Republic
[10]Finnish Meteorological Institute, Helsinki, Finland
[11]Current affiliation: Cooperative Institute for Research in Environmental Sciences, University of Colorado Boulder, USA; NOAA Earth Systems Research Laboratory, Chemical Sciences Division, Boulder, Colorado, USA
[12]School of Environment and Society, Tokyo Institute of Technology, Tokyo, Japan
[13]Deutscher Wetterdienst, Offenbach, Germany
[14]Software Consultant, Berlin, Germany
[15]Institute for Atmospheric and Earth System Research / Physics, Faculty of Science, University of Helsinki, Finland
[16]Ingenieurgesellschaft Prof. Dr. Sieker mbH, Hoppegarten, Germany
[17]Hochschule Offenburg, Offenburg, Germany
[18]Current affiliation: Oeschger Centre for Climate Change Research, University of Bern, Bern, Switzerland
[19]Current affiliation: Climate and Environmental Physics, Physics Institute, University of Bern, Bern, Switzerland
[20]Humboldt-Universität zu Berlin, Berlin, Germany
[21]Faculty of Energy Engineering, Aswan University, Aswan, Egypt
[22]Carl von Ossietzky Universität Oldenburg, ForWind - Research Center of Wind Energy, Oldenburg, Germany
[23]Current affiliation: Fraunhofer Institute for Wind Energy Systems, Oldenburg, Germany

**Correspondence:** B. Maronga (maronga@muk.uni-hannover.de)

**Abstract.** In this paper we describe the PALM model system 6.0. PALM is a Fortran based code and has been applied for studying a variety of atmospheric and oceanic boundary layers for about 20 years. The model is optimized for use on massively parallel computer architectures. This is a follow-up paper to the PALM 4.0 model description in Maronga et al. (2015). During the last years, PALM has been significantly improved and now offers a variety of new components. In particular, much effort was made to enhance the model by components needed for applications in urban environments, like fully interactive land



surface and radiation schemes, chemistry, and an indoor model. This paper serves as an overview paper of the PALM 6.0 model system and we describe its current model core. The individual components for urban applications, case studies, validation runs, and issues with suitable input data are presented and discussed in a series of companion papers in this special issue.

# 1 Introduction

Since the early 1970s, the turbulence-resolving so-called Large-eddy simulation (LES) technique has been increasingly employed for studying the atmospheric boundary layer (ABL) at large Reynolds numbers. While the earliest studies were performed at coarse grid spacings in the order of 100 m (Lilly, 1967; Deardorff, 1973), today's supercomputers are allowing for large domain runs at fine grid spacings of 1-10 m (e.g. Kanda et al., 2004; Raasch and Franke, 2011; Sullivan and Patton, 2011, among many others) or even less (Sullivan et al., 2016; Maronga and Reuder, 2017; Maronga and Bosveld, 2017). LES models solve the three-dimensional prognostic equations for momentum, temperature, humidity, and other scalar quantities (such a chemical species). The principle of LES dictates a separation of scales. Turbulence scales larger than a chosen filter width are being directly resolved by LES models, while all smaller turbulence scales are fully parameterized within a so-called sub-grid scale (SGS) model. The filter width strongly depends on the phenomenon to be studied and must be chosen in such a way that at least 90 % of the turbulence energy can be resolved (Heus et al., 2010).

In a precursor paper (Maronga et al., 2015), we gave an overview of the *Parallelized Large-eddy Simulation Model* (PALM) version 4.0. PALM is a Fortran based code and has been applied for a variety of atmospheric and oceanic boundary layers for about 20 years. The model is optimized for use on massively parallel computer architectures, but can be used in principle also on small workstations and notebooks. The model domain is discretized in space using finite differences and equidistant horizontal grid spacings. The parallelization of the code is achieved by a 2-D domain decomposition method along the $x$ and $y$ direction on a Cartesian grid with (usually) equally sized subdomains. Ghost layers are added at the side boundaries of the subdomains in order to account for the local data dependencies, which are caused by the need to compute finite differences at these positions. A Cartesian topography (complex terrain and buildings) is available in PALM, which is based on the mask method (Briscolini and Santangelo, 1989) and allows for explicitly resolving solid obstacles such as buildings and orography. PALM also has an ocean option, allowing for studying the ocean mixed layer where the sea surface is defined at the top of the model, and which includes a prognostic equation for salinity.

Furthermore, PALM has offered several embedded models which were described in the precursor paper, namely bulk cloud microphysics parameterizations, a Lagrangian particle model (LPM) which can be used for studying dispersion processes in turbulent flows, or as a Lagrangian cloud model (LCM) employing the super-droplet approach. Moreover, a plant canopy model can be used to study effects of plants as obstacles on the flow. A 1-D version of PALM can be switched on in order to generate steady-state wind profiles for 3-D model intialization.

Due to the enormous amount of data that comes along with computationally expensive LES (in terms of number of grid points and short time steps), the data handling plays a key role for the performance of LES models and for data analysis during post-processing. PALM is optimized to pursue the strategy of performing data operations like time- or domain-averaging to





great extent on-line instead of postpone such operations to a post-processing step. In this way, the data output (e.g., of huge 4-D data, or temporal averages) can be significantly reduced. In order to allow the user to perform own calculations during runtime, a user interface offers a wide range of possibilities, e.g., for defining user-defined output quantities. PALM allows data output for different quantities as time series, (horizontally-averaged) vertical profiles, 2-D cross sections, 3-D volume data, and

masked data. All data output files are in netCDF format, which can be processed by a variety of public domain and commercial software. The only exception is data output from the LPM, which is output in Fortran binary format for a better performance. For details about PALM's specifics, application scenarios, and validation runs see Maronga et al. (2015) and references therein.

In the present paper we describe the PALM model system version 6.0. Since version 4.0, the code has undergone massive changes and improvements. Above all, new components for applications of PALM in urban environments, so-called PALM-

4U (*PALM for urban applications*) components, have been added in the scope of the Urban Climate Under Change [UC]$^2$ framework funded by the German Federal Ministry of Education and Research (Scherer et al., 2019b; Maronga et al., 2019). Besides, a turbulence closure based on the Reynolds-averaged Navier-Stokes (RANS) equations was added, enabling PALM to not only run in turbulence-resolving (i.e. LES), but also in RANS mode where the full turbulence spectrum is parameterized. Originally, the name PALM referred to its parallelization as a special feature of the model. Nowadays, however, most of the

existing LES models are parallelized. Moreover, with the RANS mode implemented, PALM is more than an LES model, rendering the full name of the model inappropriate. As the name PALM has been established in the research community, we thus decided to drop the full name and use the abbreviation PALM as a proper name from now on. The model is now referred to as the PALM model system, consisting of the PALM model core and the PALM-4U components. A motivation for developing the PALM-4U components and a description of model developments done within [UC]$^2$, the reader is referred to Maronga

et al. (2019). As the model core in version 4.0 was described in detail in the precursor paper, we will here focus on the changes in the model core and give an overview of all new components that have been added to the model. The individual new PALM-4U components, case studies, validation runs, and issues with suitable input data are presented and discussed in a series of companion papers in this special issue.

The paper is organized as follows: Sect. 2 deals with the description of the model core, while Sect. 3 and Sect. 4 give details

about the embedded modules in the PALM core and the PALM-4U components, respectively. Sect. 5 provides technical details, including recent developments in model operation, data structure of surface elements, I/O data handling, and optimization. The paper closes with conclusions in Sect. 6. Note that all symbols that will be introduced in the following are also listed in Tables 1 - 8.

## 2 PALM model core

In this section, we give a detailed description of the changes of the PALM model core since version 4.0. We here confine ourselves to the atmospheric version. Details about the ocean version are given by Maronga et al. (2015) and Sect. 2.4. By default, PALM solves equations for up to seven prognostic variables: the velocity components $u, v, w$ on a staggered Cartesian grid (staggered Arakawa-C grid Harlow and Welch, 1965; Arakawa and Lamb, 1977), potential temperature $\theta$, SGS turbulence





**Table 1.** List of general model parameters.

| Symbol | Value | Description |
| --- | --- | --- |
| $c_0, c_1, c_2, c_3$ | 0.55, 1.44, 1.92, 1.44 | Model constants in RANS turbulence parameterization |
| $c_p$ | $1005\,\mathrm{J\,kg^{-1}\,K^{-1}}$ | Specific heat capacity of dry air at constant pressure |
| $g$ | $9.81\,\mathrm{m\,s^{-2}}$ | Gravitational acceleration |
| $l_\mathrm{v}$ | $2.5 \times 10^6\,\mathrm{J\,kg^{-1}}$ | Specific latent heat of vaporization |
| $Pr$ | 1 | Prandtl number in RANS turbulence parameterization |
| $p_0$ | $1000\,\mathrm{hPa}$ | Reference air pressure |
| $R_\mathrm{d}$ | $287\,\mathrm{J\,kg^{-1}\,K^{-1}}$ | Specific gas constant for dry air |
| $R_\mathrm{v}$ | $461.51\,\mathrm{J\,kg^{-1}\,K^{-1}}$ | Specific gas constant for water vapor |
| $S_0$ | $1368\,\mathrm{W\,m^{-2}}$ | Solar constant |
| $\alpha_\mathrm{Ch}$ | 0.018 | Charnock constant |
| $\epsilon_\mathrm{atm}$ | 0.8 | Atmospheric emissivity |
| $\kappa$ | 0.4 | Kármán constant |
| $\nu$ | $1.461 \times 10^{-5}\,\mathrm{m^2\,s^{-1}}$ | Kinematic viscosity of air |
| $\pi$ | $3.14159\ldots$ | Pi |
| $\sigma_e$ | 1.0 | Model constant in RANS turbulence parameterization |
| $\sigma_\epsilon$ | 1.3 | Model constant in RANS turbulence parameterization |
| $\sigma_\mathrm{SB}$ | $5.67 \cdot 10^{-8}\,\mathrm{W\,m^{-2}\,K^{-4}}$ | Stefan-Boltzmann constant |
| $\Omega$ | $0.729 \times 10^{-4}\,\mathrm{rad\,s^{-1}}$ | Angular velocity of the Earth |

kinetic energy (SGS-TKE) $e$ (in LES mode), water vapor mixing ratio $q_\mathrm{v}$, and possibly a passive scalar $s$. Note that in PALM 4.0, it was only possible to use either water vapor or passive scalar as both used the same prognostic equation in the model code, while both are now fully separated and can be used simultaneously.

## 2.1 Governing equations of the PALM core

5 By default, PALM solves incompressible approximations of the Navier-Stokes equations, either in Boussinesq-approximated form, filtered based on a spatial scale separation approach after Schumann (1975) (described in Maronga et al., 2015), or in an anelastic approximation, in which the flow is treated as incompressible, but allowing for density variations with height, while variations in time are not permitted. This enables the application of PALM to simulate atmospheric phenomena that extend throughout the entire troposphere (e.g. deep convection). Both, anelastic and Boussinesq-approximated forms are described by

10 a single set of equations that only differ in the treatment of the density $\rho$. For the Boussinesq-form $\rho$ is set to a constant value (and then drops out of most terms), while the anelastic-form results from varying $\rho$ with height during initialization.

In the following set of equations, angular brackets denote a horizontal domain average. A subscript 0 indicates a surface value. Note that the variables in the equations are implicitly filtered by the discretization (see above), but that the continuous





**Table 2.** List of general symbols.

| Symbol | Dimension | Description |
|---|---|---|
| $F$ | N | Random forcing term in parameterization of wave breaking |
| $N_{\text{chem}}$ | | Number of chemical species |
| $s$ | $\text{kg m}^{-3}$ | Passive scalar |
| $T$ | K | Absolute air temperature |
| $U_{\text{s}}$ | $\text{m s}^{-1}$ | Wave amplitude in Stokes drift parameterization |
| $u_i$ | $\text{m s}^{-1}$ | Velocity components ($u_1 = u, u_2 = v, u_3 = w$) |
| $u_{\text{g},i}$ | $\text{m s}^{-1}$ | Geostrophic wind components ($u_{\text{g},1} = u_{\text{g}}, u_{\text{g},2} = v_{\text{g}}$) |
| $u_{\text{s}}$ | $\text{m s}^{-1}$ | Stokes drift velocity |
| $u_{\text{tr}}$ | $\text{m s}^{-1}$ | Transport velocity used for radiation boundary conditions at the model outflow |
| $x_{\text{d}}$ | m | Distance in $x-$direction used for radiation boundary conditions at the model outflow |
| $x_i$ | m | Coordinate on the Cartesian grid ($x_1 = x, x_2 = y, x_3 = z$) |
| $z_{\text{w}}$ | m | Wave height in Stokes drift parameterization |
| $\Delta$ | m | Grid spacing |
| $\Delta x, \Delta y, \Delta z$ | m | Grid spacings in $x, y, z$ direction |
| $\Delta t$ | s | Time step of the LES model |
| $\delta$ | | Kronecker-delta |
| $\theta$ | K | Potential temperature |
| $\theta_{\text{v}}$ | K | Virtual potential temperature |
| $\theta_{\text{v,ref}}$ | K | Reference state of virtual potential temperature |
| $\lambda_{\text{w}}$ | m | Wavelength in Stokes drift parameterization |
| $\Pi$ | | Exner function |
| $\pi^*$ | hPa | Perturbation pressure |
| $\rho$ | $\text{kg m}^{-3}$ | Density of dry air (basic state) |
| $\rho_\theta$ | $\text{kg m}^{-3}$ | Potential density |
| $\omega$ | $\text{s}^{-1}$ | Rotation of velocity |

form of the equations is used here for convenience. A double prime indicates SGS variables. The overbar indicating filtered quantities is omitted for readability, except for the SGS flux terms. The equations for the conservation of mass, momentum, thermal internal energy, moisture and another arbitrary passive scalar quantity, filtered over a grid volume on a Cartesian grid, then read as

$$5 \quad \frac{\partial u_j \rho}{\partial x_j} = 0 \qquad (1)$$





$$\frac{\partial u_i}{\partial t} = -\frac{1}{\rho}\frac{\partial \rho u_i u_j}{\partial x_j} - \varepsilon_{ijk} f_j u_k + \varepsilon_{i3j} f_3 u_{\mathrm{g},j} - \frac{\partial}{\partial x_i}\left(\frac{\pi^*}{\rho}\right) \tag{2}$$
$$+ g\frac{\theta_\mathrm{v} - \theta_\mathrm{v,ref}}{\theta_\mathrm{v,ref}}\delta_{i3} - \frac{1}{\rho}\frac{\partial}{\partial x_j}\rho\left(\overline{u_i'' u_j''} - \frac{2}{3}e\delta_{ij}\right),$$

$$\frac{\partial \theta}{\partial t} = -\frac{1}{\rho}\frac{\partial \rho u_j \theta}{\partial x_j} - \frac{1}{\rho}\frac{\partial}{\partial x_j}\left(\rho\overline{u_j'' \theta''}\right) - \frac{l_\mathrm{v}}{c_p \Pi}\Psi_{q_\mathrm{v}} \tag{3}$$

$$\frac{\partial q_\mathrm{v}}{\partial t} = -\frac{1}{\rho}\frac{\partial \rho u_j q_\mathrm{v}}{\partial x_j} - \frac{1}{\rho}\frac{\partial}{\partial x_j}\left(\rho\overline{u_j'' q_\mathrm{v}''}\right) + \Psi_{q_\mathrm{v}} \tag{4}$$

$$\frac{\partial s}{\partial t} = -\frac{1}{\rho}\frac{\partial \rho u_j s}{\partial x_j} - \frac{1}{\rho}\frac{\partial}{\partial x_j}\left(\rho\overline{u_j'' s''}\right) + \Psi_s. \tag{5}$$

Here, $i,j,k \in \{1,2,3\}$. $u_i$ are the velocity components ($u_1 = u, u_2 = v, u_3 = w$) with location $x_i$ ($x_1 = x, x_2 = y, x_3 = z$), $t$ is time, $f_i = (0, 2\Omega\cos(\phi), 2\Omega\sin(\phi))$ is the Coriolis parameter with $\Omega = 0.729 \times 10^{-4}\,\mathrm{rad\,s^{-1}}$ being the Earth's angular velocity and $\phi$ being the geographical latitude. $u_{\mathrm{g},j}$ are the geostrophic wind speed components, $\rho$ is the basic state density of dry air, $\pi^* = p^* + \frac{2}{3}\rho e$ is the modified perturbation pressure with $p^*$ being the perturbation pressure and $e = \frac{1}{2}\overline{u_i'' u_i''}$, $g = 9.81\,\mathrm{m\,s^{-2}}$ is the gravitational acceleration, $\delta$ is the Kronecker delta, and $l_\mathrm{v} = 2.5 \times 10^6\,\mathrm{J\,kg^{-1}}$ is the specific latent heat of vaporization. The reference state $\theta_\mathrm{v,ref}$ in Eq. (2) can be set to be the horizontal average $\langle\theta_\mathrm{v}\rangle$, the initial state, or a fixed reference value. Furthermore, $\Psi_{q_\mathrm{v}}$ and $\Psi_s$ are source/sink terms of $q_\mathrm{v}$ and $s$, respectively. The potential temperature is defined as

$$\theta = T/\Pi, \tag{6}$$

with the absolute temperature $T$ and the Exner function

$$\Pi = \left(\frac{p}{p_0}\right)^{R_\mathrm{d}/c_p}, \tag{7}$$

with $p$ being the hydrostatic air pressure, $p_0 = 1000\,\mathrm{hPa}$ a reference pressure, $R_\mathrm{d} = 287\,\mathrm{J\,kg^{-1}K^{-1}}$ the specific gas constant for dry air, and $c_p = 1005\,\mathrm{J\,kg^{-1}K^{-1}}$ the specific heat of dry air at constant pressure. The virtual potential temperature is defined as

$$\theta_\mathrm{v} = \theta\left[1 + \left(\frac{R_\mathrm{v}}{R_\mathrm{d}} - 1\right)q_\mathrm{v} - q_\mathrm{l}\right] \tag{8}$$

with the specific gas constant for water vapor $R_\mathrm{v} = 461.51\,\mathrm{J\,kg^{-1}\,K^{-1}}$, and the liquid water mixing ratio $q_\mathrm{l}$. For the computation of $q_\mathrm{l}$, see the descriptions of the embedded cloud microphysical models in Sects. 3.1 and 3.4.

## 2.2 Turbulence closures

By default, PALM employs a 1.5-order closure (LES mode) after Deardorff (1980) in the formulation by Moeng and Wyngaard (1988) and Saiki et al. (2000) (hereafter referred to as Deardorff scheme). Details are given in Maronga et al. (2015). Since version 6.0, an alternative dynamic SGS closure can be used which will be described in the following. Moreover, two turbulence closures are available in RANS mode (i.e. the full spectrum of turbulence is parameterized): a so-called TKE-$l$ and a TKE-$\epsilon$ closure, where $l$ is a mixing length and $\epsilon$ is the SGS-TKE dissipation rate.





**Table 3.** List of SGS model symbols.

| Symbol | Dimension | Description |
| --- | --- | --- |
| $c_*$ | | Dynamic subgrid-scale coefficient |
| $e$ | $\mathrm{m^2\,s^{-2}}$ | Subgrid-scale turbulence kinetic energy (total turbulent kinetic energy in RANS mode) |
| $l$ | m | Mixing length |
| $l_\mathrm{B}$ | m | Mixing length after Blackadar (1962) |
| $l_\mathrm{wall}$ | m | Minimum mixing length |
| $K_\mathrm{h}$ | $\mathrm{m^2\,s^{-1}}$ | SGS eddy diffusivity of heat |
| $K_\mathrm{m}$ | $\mathrm{m^2\,s^{-1}}$ | SGS eddy diffusivity of momentum |
| $L_{ij}$ | $\mathrm{m^2 s^{-2}}$ | Resolved stress tensor |
| $S_{ij}$ | $\mathrm{s^{-1}}$ | Strain tensor |
| $T_{ij}$ | $\mathrm{m^2 s^{-2}}$ | Subtest-scale stress tensor |
| $\epsilon$ | $\mathrm{m^2\,s^{-3}}$ | SGS-TKE dissipation rate |
| $\nu_*^\mathrm{T}$ | $\mathrm{m^2 s}$ | Subtest-scale viscosity parameter |
| $\tau_{ij}$ | $\mathrm{m^2 s^{-2}}$ | SGS stress tensor |
| $\tau_{\mathrm{d},ij}$ | $\mathrm{m^2 s^{-2}}$ | Deviatoric SGS stress tensor |

### 2.2.1 Dynamic SGS closure

The dynamic SGS closure follows Heinz (2008) and Mokhtarpoor and Heinz (2017). In general, the dynamic SGS closure employs the same equations for calculating the SGS fluxes as the Deardorff scheme, assuming that the energy transport by SGS eddies is proportional to the local gradients of the mean resolved quantities and reads

$$\overline{u_i'' u_j''} - \frac{2}{3} e \delta_{ij} = -K_\mathrm{m} \left( \frac{\partial \overline{u}_i}{\partial x_j} + \frac{\partial \overline{u}_j}{\partial x_i} \right) \tag{9}$$

$$\overline{u_i'' \theta''} = -K_\mathrm{h} \frac{\partial \overline{\theta}}{\partial x_i} \tag{10}$$

$$\overline{u_i'' q_\mathrm{v}''} = -K_\mathrm{h} \frac{\partial \overline{q}_\mathrm{v}}{\partial x_i} \tag{11}$$

$$\overline{u_i'' s''} = -K_\mathrm{h} \frac{\partial \overline{s}}{\partial x_i} \tag{12}$$

where $K_\mathrm{m}$ and $K_\mathrm{h}$ are the local SGS diffusivities of momentum and heat, respectively. In order to distinguish between different filter operations the overbar is used to denote variables that are filtered with the horizontal grid spacing $\boldsymbol{\Delta}$ in this subsection. While $K_\mathrm{h}$ is calculated as in the Deardorff scheme, a dynamic approach is applied to calculate $K_\mathrm{m}$, viz.

$$K_\mathrm{m} = c_* \, \boldsymbol{\Delta}_\mathrm{max} \, \sqrt{e}, \tag{13}$$




where $\boldsymbol{\Delta}_{\mathrm{max}} = \max(\Delta_{\mathrm{x}}, \Delta_{\mathrm{y}}, \Delta_{\mathrm{z}})$. Unlike in the Deardorff scheme $c_*$ is not a fixed value but is calculated at each timestep for each grid cell. As for the Deardorff scheme, $e$ is calculated using a prognostic equation:

$$\frac{\partial e}{\partial t} = -u_j \frac{\partial e}{\partial x_j} - \left(\overline{u_i'' u_j''}\right) \frac{\partial \overline{u}_i}{\partial x_j} + \frac{g}{\theta_{\mathrm{v,ref}}} \overline{u_3'' \theta''} - \frac{\partial}{\partial x_j}\left[\overline{u_j''\left(e + \frac{p''}{\rho}\right)}\right] - (0.19 + 0.74l/\boldsymbol{\Delta})\frac{e^{3/2}}{l}, \tag{14}$$

with $l$ being a mixing length. Note that in the SGS closures, $\theta_{\mathrm{v,ref}}$ refers to either a given reference value or the local value of $\theta$. The pressure term in Eq. (14) is parameterized as

$$\overline{\left[u_j''\left(e + \frac{p''}{\rho}\right)\right]} = -2K_m \frac{\partial e}{\partial x_j}. \tag{15}$$

The left hand side of Eq. (9) is called deviatoric subgrid stress. Using the rate of strain tensor $S_{ij} = 0.5\left(\frac{\partial u_i}{\partial x_j} + \frac{\partial u_j}{\partial x_i}\right)$ it can be written as follows:

$$\tau_{ij}^d = \tau_{ij} - \frac{\tau_{nn}}{3}\delta_{ij} = -2K_{\mathrm{m}}\overline{S}_{ij}, \tag{16}$$

where we used the summation convention. The subgrid stress can also be expressed as $\tau_{ij} = \overline{u_i u_j} - \overline{u}_i \overline{u}_j$. This expression makes clear why the subgrid stress has to be modelled, since only the second term of the right hand side is known. Following Germano et al. (1991), a test filter is introduced, which is $\boldsymbol{\Delta}^{\mathrm{T}} = 2\boldsymbol{\Delta}$ in our case. The subgrid stress on the test filter scale then is $T_{ij} = \widehat{\overline{u_i u_j}} - \widehat{\overline{u}}_i\widehat{\overline{u}}_j$, where also the first term on the right hand side is unknown (the hat denotes a filter operation with the width of the test filter). The difference between subgrid stress on the test filter level and the test filtered subgrid stress is the resolved stress $L_{ij} = T_{ij} - \widehat{\tau}_{ij} = \widehat{\overline{u}_i\overline{u}_j} - \widehat{\overline{u}}_i\widehat{\overline{u}}_j$. Both terms on the right hand side are known and $L_{ij}$ can thus be calculated directly by application of the test filter to the resolved velocities on the grid cells. As described in Heinz (2008), $c_*$ can be calculated via

$$c_* = -\frac{L_{ij}^d \widehat{\overline{S}}_{ji}}{2\nu_*^{\mathrm{T}}\widehat{\overline{S}}_{mn}\widehat{\overline{S}}_{nm}}, \tag{17}$$

where $\nu_*^{\mathrm{T}} = \boldsymbol{\Delta}^{\mathrm{T}}\left(L_{ii}/2\right)^2$ is the subtest-scale viscosity. The stability of the simulation is ensured by using dynamic bounds that keep the values of $c_*$ in the range

$$|c_*| \leq \frac{23}{24\sqrt{3}}\frac{\sqrt{e}}{\boldsymbol{\Delta}\sqrt{\overline{S}_{ij}\overline{S}_{ji}}}, \tag{18}$$

as derived by Mokhtarpoor and Heinz (2017). This model does not need artificial limitation of the range of $c_*$ for stable runs and allows the occurence of energy backscatter (i.e., negative values of $K_{\mathrm{m}}$). Unlike other dynamic models, this formulation of $c_*$ is not derived using model assumptions for the subgrid stress and the stress on the test filter level, but is derived as consequence of stochastic analysis (Heinz, 2008; Heinz and Gopalan, 2012).

### 2.2.2 RANS turbulence closures

For RANS mode, PALM offers two different turbulence closures, a TKE-$l$ and the standard TKE-$\epsilon$ closure (Mellor and Yamada, 1974, 1982), to calculate the eddy diffusivities, which then describe diffusion by the complete turbulence spectrum. While





the TKE-$l$ closure uses a single prognostic equation to calculate the TKE, the standard TKE-$\epsilon$ closure applies an additional prognostic equation for $\epsilon$ in addition to the equation for $e$.

In the TKE-$l$ closure (e.g. Holt and Raman, 1988), the eddy diffusivities are calculated via $e$ and $l$ as

$$K_{\mathrm{m}} = c_0 l \sqrt{e}, \tag{19}$$

$$K_{\mathrm{h}} = \frac{K_{\mathrm{m}}}{Pr}, \tag{20}$$

where $Pr = 1$ denotes the Prandtl number and $c_0 = 0.55$ a model constant. The Prandtl number can be adjusted by the user for different stability regimes. Note that, in case of RANS mode, $e$ denotes the total turbulent kinetic energy as the full turbulence spectrum is parameterized. To calculate $e$, Eq. (14) is modified by introducing gradient approaches for the turbulent transport terms:

$$\frac{\partial e}{\partial t} = -u_j \frac{\partial e}{\partial x_j} + K_{\mathrm{m}} \left( \frac{\partial u_i}{\partial x_j} + \frac{\partial u_j}{\partial x_i} \right) \frac{\partial u_i}{\partial x_j} - \frac{g}{\theta_{\mathrm{v,ref}}} K_{\mathrm{h}} \frac{\partial \theta_{\mathrm{v,ref}}}{\partial z} + K_{\mathrm{e}} \frac{\partial^2 e}{\partial x_j^2} - \epsilon. \tag{21}$$

Here, $K_{\mathrm{e}} = \dfrac{K_{\mathrm{m}}}{\sigma_e}$ is the diffusivity of $e$, with the model constant $\sigma_e = 1$ as default value, and $\epsilon$ is calculated as

$$\epsilon = c_0^3 e \frac{\sqrt{e}}{l}. \tag{22}$$

The mixing length $l$ is calculated using the mixing length after Blackadar (1962) $l_{\mathrm{B}}$ and the similarity function of momentum $\Phi_{\mathrm{m}}$ for stable conditions in the formulation of Businger–Dyer (see, e.g., Panofsky and Dutton, 1984):

$$l = \begin{cases} \min\left( \dfrac{l_{\mathrm{B}}}{\Phi_{\mathrm{m}}}, l_{\mathrm{wall}} \right) & \text{for } \frac{z}{L} \geq 0, \\ \min\left( l_{\mathrm{B}}, l_{\mathrm{wall}} \right) & \text{for } \frac{z}{L} < 0, \end{cases} \tag{23}$$

with

$$l_{\mathrm{B}} = \frac{\kappa z}{1 + \frac{\kappa z}{0.00027 \left( u_{\mathrm{g},1}^2 + u_{\mathrm{g},2}^2 \right)^{0.5} f}}, \text{ and} \tag{24}$$

$$\Phi_{\mathrm{m}} = 1 + 5 \frac{z}{L}, \tag{25}$$

where $\kappa = 0.4$ denotes the Von Kármán constant, $L$ the Obukhov length, and $z$ the height above the surface. The mixing length is limited by $l_{\mathrm{wall}}$, which is the distance to the nearest solid surface.

Aside from the TKE-$l$ closure, also a standard TKE-$\epsilon$ model is available as a turbulence closure. When choosing the standard TKE-$\epsilon$ model, $K_{\mathrm{m}}$ is calculated via

$$K_{\mathrm{m}} = c_0^4 \frac{e^2}{\epsilon}. \tag{26}$$

The modeled TKE is calculated using Eq. (21) and an additional prognostic equation is used to calculate $\epsilon$:

$$\frac{\partial \epsilon}{\partial t} = -u_j \frac{\partial \epsilon}{\partial x_j} + c_1 \frac{\epsilon}{e} K_{\mathrm{m}} \left( \frac{\partial u_i}{\partial x_j} + \frac{\partial u_j}{\partial x_i} \right) \frac{\partial u_i}{\partial x_j} - c_3 \frac{\epsilon}{e} \frac{g}{\theta_{\mathrm{v,ref}}} K_{\mathrm{h}} \frac{\partial \theta_{\mathrm{v,ref}}}{\partial z} + K_\epsilon \frac{\partial^2 \epsilon}{\partial x_j^2} - c_2 \frac{\epsilon^2}{e}, \tag{27}$$





where $K_\epsilon = \frac{K_\mathrm{m}}{\sigma_\epsilon}$ with $\sigma_\epsilon = 1.3$ and $c_1 = 1.44$, $c_2 = 1.92$, and $c_3 = 1.44$ are being model constants (e.g. Launder and Spalding, 1974; Oliveira and Younis, 2000). As the constants $c_0 - c_3$ as well as $\sigma_e$ and $\sigma_\epsilon$ depend on the situation studied, they might need to be adjusted by the user.

For a validation of the RANS turbulence parameteriztion in PALM 6.0 against LES results and wind tunnel experiments, see Gronemeier et al. (2019, to be submitted to this special issue).

### 2.3 Boundary conditions

#### 2.3.1 Constant flux layer

Following Monin-Obukhov similarity theory (MOST), a constant flux layer assumption is used between the surface and the first computational grid level ($k = 1$, $z_\mathrm{mo} = 0.5 \cdot \Delta z$). Using roughness lengths for heat, humidity, and momentum ($z_{0,\mathrm{h}}$, $z_{0,\mathrm{q}}$, and $z_0$, respectively), MOST then provides surface fluxes of momentum (shear stress) and scalar quantities (heat and moisture flux) as bottom boundary condition. In PALM it is assumed that MOST can be applied locally, even though there is no theoretical foundation for this assumption. Hultmark et al. (2013), e.g., pointed out that this leads to a systematical overprediction of the mean shear stress. However, this local method has the advantage that surface heterogeneities can be prescribed at the surface; and therefore it has become standard in most contemporary LES codes.

The surface layer vertical profile of the horizontal wind velocity $u_\mathrm{h} = (u^2 + v^2)^{\frac{1}{2}}$ is predicted by MOST through

$$\frac{\partial u_\mathrm{h}}{\partial z} = \frac{u_*}{\kappa z} \Phi_\mathrm{m} \left( \frac{z}{L} \right) , \tag{28}$$

where $\Phi_\mathrm{m}$ is the similarity function for momentum in the formulation of Businger–Dyer (see e.g. Panofsky and Dutton, 1984):

$$\Phi_\mathrm{m} = \begin{cases} 1 + 5\frac{z}{L} & \text{for } \frac{z}{L} \geq 0 , \\ \left(1 - 16\frac{z}{L}\right)^{-\frac{1}{4}} & \text{for } \frac{z}{L} < 0 . \end{cases} \tag{29}$$

The scaling parameters $\theta_*$ and $q_*$ are defined by MOST as:

$$\theta_* = -\frac{\overline{w''\theta''}_0}{u_*}, \; q_* = -\frac{\overline{w''q_\mathrm{v}''}_0}{u_*} , \tag{30}$$

with the friction velocity $u_*$ (defined through the square root of the surface shear stress) as

$$u_* = \left[ \left( \overline{u''w''}_0 \right)^2 + \left( \overline{v''w''}_0 \right)^2 \right]^{\frac{1}{4}} . \tag{31}$$

In PALM, $u_*$ is calculated from $u_\mathrm{h}$ at $z_\mathrm{mo}$ by vertical integration of Eq. (28) over $z$ from $z_0$ to $z_\mathrm{mo}$.

From Eqs. (28) and (31) it is possible to derive a formulation for the horizontal wind components, viz.

$$\frac{\partial u}{\partial z} = \frac{-\overline{u''w''}_0}{u_* \kappa z} \Phi_\mathrm{m} \left( \frac{z}{L} \right) \quad \text{and} \quad \frac{\partial v}{\partial z} = \frac{-\overline{v''w''}_0}{u_* \kappa z} \Phi_\mathrm{m} \left( \frac{z}{L} \right) . \tag{32}$$

Vertical integration of Eq. (32) over $z$ from $z_0$ to $z_\mathrm{mo}$ then yields the surface momentum fluxes $\overline{u''w''}_0$ and $\overline{v''w''}_0$.





The formulations above all require knowledge of the scaling parameters $\theta_*$ and $q_*$. These are deduced from vertical integration of

$$\frac{\partial \theta}{\partial z} = \frac{\theta_*}{\kappa z}\Phi_{\mathrm{h}}\left(\frac{z}{L}\right) \quad \text{and} \quad \frac{\partial q_{\mathrm{v}}}{\partial z} = \frac{q_*}{\kappa z}\Phi_{\mathrm{h}}\left(\frac{z}{L}\right) \tag{33}$$

over $z$ from $z_{0,\mathrm{h}}$ to $z_{\mathrm{mo}}$. The similarity function $\Phi_{\mathrm{h}}$ is given by

$$\Phi_{\mathrm{h}} = \begin{cases} 1 + 5\frac{z}{L} & \text{for } \frac{z}{L} \geq 0, \\ \left(1 - 16\frac{z}{L}\right)^{-1/2} & \text{for } \frac{z}{L} < 0. \end{cases} \tag{34}$$

Previously, the implementation of the constant flux layer involved a diagnostic-prognostic equation for $L$, based on data from the previous time step. Even though it was found that this method introduces only negligible errors, we decided to revise this procedure and calculate $L$ based on using a Newton iteration method instead. By doing so, we can achieve a correct value of $L$ which can be important when the model is coupled to a surface scheme. We also found that this does not increase the

computational costs to a significant amount (usually less than 1 %). Since PALM 6.0 (revision 3668) Newton iteration is the only available method. The Newton iteration method involves the calculation of a bulk Richardson number $Ri_{\mathrm{b}}$. Depending on whether fluxes are prescribed or Dirichlet boundary conditions are used for temperature and humidity, $Ri_{\mathrm{b}}$ is related to $L$ via

$$Ri_{\mathrm{b}} = \frac{z_{\mathrm{mo}}}{L} \cdot \begin{cases} \dfrac{\phi_{\mathrm{H}}}{\phi_{\mathrm{M}}^2} & \text{for Dirichlet conditions,} \\ \dfrac{1}{\phi_{\mathrm{M}}^3} & \text{for prescribed fluxes,} \end{cases} \tag{35}$$

where

$$\phi_{\mathrm{H}} = \log\left(\frac{z_{\mathrm{mo}}}{z_{0,\mathrm{h}}}\right) - \Phi_{\mathrm{H}}\left(\frac{z_{\mathrm{mo}}}{L}\right) + \Phi_{\mathrm{H}}\left(\frac{z_{0,\mathrm{h}}}{L}\right), \tag{36}$$

and

$$\phi_{\mathrm{M}} = \log\left(\frac{z_{\mathrm{mo}}}{z_0}\right) - \Phi_{\mathrm{M}}\left(\frac{z_{\mathrm{mo}}}{L}\right) + \Phi_{\mathrm{M}}\left(\frac{z_{0,\mathrm{h}}}{L}\right), \tag{37}$$

are the integrated universal profile stability functions of $\Phi_{\mathrm{M}}$ and $\Phi_{\mathrm{H}}$ (see Paulson, 1970; Holtslag and De Bruin, 1988), so that a (bulk) Richardson number can be defined:

$$Ri_{\mathrm{b}} = \begin{cases} \dfrac{gz_{\mathrm{mo}}(\theta_{\mathrm{v,mo}} - \theta_{\mathrm{v,0}})}{u_{\mathrm{h}}^2 \theta_v} & \text{for Dirichlet conditions,} \\ -\dfrac{gz_{\mathrm{mo}}\overline{w''\theta_{\mathrm{v}0}''}}{\kappa^2 u_{\mathrm{h}}^3 \theta_v} & \text{for prescribed fluxes.} \end{cases} \tag{38}$$

The above equations are solved for $L$ by finding the root of the function $f_{\mathrm{N}}$:

$$f_{\mathrm{N}} = Ri_{\mathrm{b}} - \frac{z_{\mathrm{mo}}}{L} \cdot \begin{cases} \dfrac{[\phi_{\mathrm{H}}]}{[\phi_{\mathrm{M}}]^2} & \text{for Dirichlet conditions.} \\ \dfrac{[\phi_{\mathrm{H}}]}{[\phi_{\mathrm{M}}]^3} & \text{for prescribed fluxes.} \end{cases} \tag{39}$$





The solution is then given by iteration of

$$L^{n+1} = L^n - \frac{f_{\mathrm{N}}(L^n)}{f'_{\mathrm{N}}(L^n)} \tag{40}$$

with iteration step $n$, and

$$f'_{\mathrm{N}}(L) = \frac{df_{\mathrm{N}}}{dL} \tag{41}$$

until $L$ meets a convergence criterion.

The surface fluxes of sensible and latent heat, as well as the surface shear stress are then calculated using Eqs. (30) and (31). Note that for vertically oriented surfaces in combination with an interactive surface model switched on (see Sects. 3.5 and 4.5), the surface fluxes are calculated after Krayenhoff and Voogt (2007) as static stability considerations do not apply for such surface orientations (see also Resler et al., 2017).

In case of the TKE-$\epsilon$ RANS closure, the boundary condition for $e$, $\epsilon$, and $K_{\mathrm{m}}$ are

$$e = \left( \frac{u_*}{c_0} \right), \tag{42}$$

$$\epsilon = \frac{u_*^3}{\kappa z_{\mathrm{mo}}}, \tag{43}$$

$$K_{\mathrm{m}} = \kappa u_* z_{\mathrm{mo}} \Phi_{\mathrm{m}}^{-1} \left( \frac{z_{\mathrm{mo}}}{L} \right). \tag{44}$$

### 2.3.2   Wave-depending surface roughness

As the ocean surface in PALM is assumed to be flat and waves are not explicitly resolved, a Charnock parameterization can be switched on which relates the surface roughness lengths to the friction velocity as described in Beljaars (1994). This accounts for the fact that water surfaces become aerodynamically smooth for low wind speeds. For ocean surfaces, the roughness lengths are thus calculated for each surface grid point as

$$z_0 = \frac{0.11\nu}{u_*} + \alpha_{\mathrm{Ch}} \frac{u_*^2}{g}, \tag{45}$$

$$z_{0,\mathrm{h}} = \frac{0.4\nu}{u_*}, \tag{46}$$

$$z_{0,\mathrm{q}} = \frac{0.62\nu}{u_*} \tag{47}$$

with $\alpha_{\mathrm{Ch}} = 0.0018$ being the Charnock constant, and $\nu = 1.461 \times 10^{-5}\,\mathrm{m}^2\,\mathrm{s}^{-1}$ being the kinematic viscosity. Note that this parameterization is designed for large-scale models where waves are a sub-grid scale phenomenon. For fine grid spacings and/or large waves (in amplitude and wavelength), this parameterization can lead to erroneous roughness lengths and should

not be switched on without rigorous testing.

### 2.3.3   Lateral boundary conditions

At lateral domain boundaries, various different conditions can be applied which are listed in Tab. 9.





**Table 4.** List of surface layer symbols.

| Symbol | Dimension | Description |
| --- | --- | --- |
| $L$ | m | Obukhov length |
| $q_*$ | $\mathrm{kg\,kg^{-1}}$ | MOST humidity scale |
| $Ri_\mathrm{b}$ | | Bulk Richardson number |
| $u_\mathrm{h}$ | $\mathrm{m\,s^{-1}}$ | Absolute value of the horizontal wind |
| $z_\mathrm{mo}$ | m | Height above the surface where MOST is applied |
| $z_0$ | m | Roughness length for momentum |
| $z_\mathrm{0,h}$ | m | Roughness length for heat |
| $z_\mathrm{0,q}$ | m | Roughness length for moisture |
| $\theta_*$ | K | MOST temperature scale |
| $\Phi_\mathrm{h}$ | | Similarity function for heat |
| $\Phi_\mathrm{m}$ | | Similarity function for momentum |
| $\Phi_\mathrm{H}$ | | Integrated similarity functions for heat |
| $\Phi_\mathrm{M}$ | | Integrated similarity functions for momentum |
| $\phi_\mathrm{H}$ | | Integrated similarity functions term for heat |
| $\phi_\mathrm{M}$ | | Integrated similarity functions term for momentum |

By default, cyclic boundary conditions apply at all lateral domain boundaries. Choosing an inflow boundary condition at one of the four domain boundaries requires to set an outflow condition at the opposing boundary while keeping the boundaries in perpendicular direction cyclic. An exception is made in case of model nesting, where inflow/outflow boundary conditions are set dynamically for each individual boundary grid point (see Sect. 4.8 and 4.9).

5     The simplest inflow condition is a purely laminar inflow using Dirichlet conditions at either domain boundary. A more sophisticated approach with fully developed turbulence already present at the inflow boundary can be achieved by using the turbulence-recycling method, which is implemented according to Lund et al. (1998) and Kataoka and Mizuno (2002). The turbulence-recycling method sets a fixed mean inflow condition at one side of the simulation domain and adds a turbulent signal from within the model domain to these mean profiles. This then creates a turbulent inflow (see Maronga et al., 2015). The turbulence-recycling method is currently only available at the left domain boundary, i.e., at $x = 0$.

    The downside of the turbulence-recycling method is the requirement of an additional recycling area within the model domain which is purely needed to generate turbulence and can not be used for data evaluation of the studied phenomenon. To avoid the necessity of including an additional recycling area within the simulation domain, a synthetic turbulence generator can be used instead of the turbulence-recycling method at the inflow boundary (Gronemeier et al., 2015). This turbulence generator is

15   based on the method published by Xie and Castro (2008) with the modification of Kim et al. (2013) for divergence-free inflow. The turbulence-generation method calculates stochastic fluctuations from an arrayed random number. This is realized via given length scales that are added to the mean inflow profiles using a Lund rotation (Lund et al., 1998), and a given Reynolds stress





tensor. In order to use this synthetic turbulence generator, it is thus required to provide all turbulent length scales for all three velocity components and the Reynolds stress tensor. This information can either be obtained from a different simulation or from measurements. Similarity assumptions in the boundary layer then reduce the number of prescribed parameters (i.e Reynolds stress tensors and turbulence length scales) to identical velocity and length scales (e.g. Marusic and Hutchins, 2008; Marusic

et al., 2010; Takimoto et al., 2013; Yagi et al., 2017). In combination with the offline nesting (see Sect. 4.9), PALM also offers the possibility to calculate length scales and the Reynolds stress tensor itself following the approach of Rotach et al. (1996).

At the outflow boundary, radiation conditions are used by default for the velocity components as proposed by Orlanski (1976). Velocity components are advected by a transport velocity $u_{\mathrm{tr}}$ which is calculated from the gradients of the transported velocity components normal to the boundary at the grid points next to the outflow boundary (see also Maronga et al. (2015)).

The transport velocity is restricted to $0 \leq u_{\mathrm{tr}} \leq \Delta/\Delta t$, where $\Delta t$ denotes the time-step.

In cases with weak background wind in a convective boundary layer, it was found that using the radiation condition can lead to instabilities and strong self-intensifying inflow regimes at the outflow boundary (Gronemeier et al., 2017). In order to prevent such artificial inflow situations at the outflow boundary, an empirical approach can be used at the outflow boundary, the so called turbulent outflow condition (Gronemeier et al., 2017). Instead of transporting the velocity components via the

radiation condition, instantaneous values of $u$, $v$, $w$, $\theta$, and $e$ are taken from a vertical plane situated at a distance $x_{\mathrm{d}}$ from the outflow boundary which are then mapped to the outflow boundary. By taking the information of the flow field from within the domain, occurring inflow regimes are disturbed and cannot intensify themselves as long as a proper $x_{\mathrm{d}}$ is chosen which needs to be a fair distance away from the outflow boundary. Note that the turbulent outflow condition can be transformed into the radiation condition where $u_{\mathrm{tr}} = \Delta/\Delta t$ if $x_{\mathrm{d}} = 0$. As for now, the turbulent outflow condition is only available at the right

domain boundary.

## 2.4  Ocean option

PALM's ocean option has been extended to include wave effects to account for the Langmuir circulation, which can be option-ally switched on. For this, the momentum equation is modified by including a vortex force and an additional advection by the Stokes drift following the theory by Craik and Leibovich (1976), similarly to McWilliams et al. (1997) and Skyllingstad and

Denbo (1995). Furthermore, a simple parameterization of wave breaking effects has been included. The modified momentum equations for the ocean then reads

$$\frac{\partial u_i}{\partial t} = -(u_j + u_{\mathrm{s},j})\frac{\partial u_i}{\partial x_j} - \varepsilon_{ijk}f_j(u_k + u_{\mathrm{s},k}) + \varepsilon_{i3j}f_3 u_{\mathrm{g},j} - \frac{\partial \pi^*}{\partial x_i} + \varepsilon_{ijk}u_{\mathrm{s},j}\omega_k \tag{48}$$
$$- g\frac{\rho_\theta - \langle\rho_\theta\rangle}{\langle\rho_\theta\rangle}\delta_{i3} - \frac{\partial}{\partial x_j}\left(\overline{u_i'' u_j''} - \frac{2}{3}e\delta_{ij}\right) + F_i,$$

where $u_{\mathrm{s}}$ is the Stokes drift velocity, $\rho_\theta$ the potential density, and $\omega_i = \varepsilon_{ijk}\dfrac{\partial u_k}{\partial x_j}$ the rotation of the velocity field. $F$ is a

random forcing term that represents the generation of small-scale turbulence by wave breaking. It should be kept in mind that the incompressibility assumption is used in the ocean option. It is assumed that wind stress and wave fields are in the same direction, and that the wave field is steady and monochromatic. The magnitude of the Stokes velocity along the wind stress





direction is then given by

$$u_s = U_s \exp\left(\frac{4\pi z}{\lambda_{\mathrm{w}}}\right) \tag{49}$$

with $U_s = (\pi z_{\mathrm{w}}/\lambda)^2 (g\lambda_{\mathrm{w}}/2\pi)^{1/2}$, where $z_{\mathrm{w}}$ is the wave height and $\lambda_{\mathrm{w}}$ is the wavelength. The current implementation of wave effects strictly follows Noh et al. (2004), in particular the parameterization of wave breaking. See Noh et al. (2004) for further

details, especially concerning the parameterization of the wave breaking effect. Note that Noh et al. (2004) used an earlier version of PALM, where the programming of the wave effects were completely realized via PALM's user interface.

As part of the general code modularization effort, all ocean related code has been put into one Fortran module, and a separate namelist has been created containing all ocean related steering parameters.

## 3 Embedded models

In this section, we first describe major revisions of the embedded models in the PALM core, namely in the bulk cloud microphysics parameterization (Sect. 3.1) and in the Lagrangian particle model (Sect 3.3-3.4). Subsequently, we introduce three new embedded models in PALM 6.0: a fully interactive land surface model (LSM, Sect. 3.5), which can be coupled to two different radiation models (Sect. 3.6), and a parameterization scheme for taking into account the effect of wind turbines (Sect. 3.7).

### 3.1 Bulk cloud microphysics improvements

In PALM 4.0, the bulk liquid-phase (i.e., no ice) two-moment microphysics scheme of Seifert and Beheng (2001; 2006) was implemented, which only predicts the rain droplet number concentration ($n_{\mathrm{r}}$) and rain water mixing ($q_{\mathrm{r}}$). This was extended by additional prognostic equations for the cloud droplet number concentration ($n_{\mathrm{c}}$) and the cloud water mixing ratio ($q_{\mathrm{c}}$), instead of using a fixed value for $n_{\mathrm{c}}$ and only diagnostically calculated values for $q_{\mathrm{c}}$. The additional prognostic equations are thus given by

$$\frac{\partial n_{\mathrm{c}}}{\partial t} = -u_j \frac{\partial n_{\mathrm{c}}}{\partial x_j} - \frac{\partial}{\partial x_j}\left(\overline{u_j'' n_{\mathrm{c}}''}\right) + \Psi_{n_{\mathrm{c}}}, \tag{50}$$

$$\frac{\partial q_{\mathrm{c}}}{\partial t} = -u_j \frac{\partial q_{\mathrm{c}}}{\partial x_j} - \frac{\partial}{\partial x_j}\left(\overline{u_j'' q_{\mathrm{c}}''}\right) + \Psi_{q_{\mathrm{c}}}, \tag{51}$$

with the sink/source terms for $\Psi_{n_{\mathrm{c}}}$ and $\Psi_{q_{\mathrm{c}}}$ and the SGS fluxes

$$\overline{u_j'' n_{\mathrm{c}}''} = -K_{\mathrm{h}} \frac{\partial n_{\mathrm{c}}}{\partial x_i}, \tag{52}$$

$$\overline{u_j'' q_{\mathrm{c}}''} = -K_{\mathrm{h}} \frac{\partial q_{\mathrm{c}}}{\partial x_i}. \tag{53}$$

The sink and source terms for $n_{\mathrm{c}}$ and $q_{\mathrm{c}}$ include the same microphysical processes as described by Maronga et al. (2015), namely autoconversion, accretion and sedimentation of cloud droplets as well as activation and diffusional growth, which has





been newly added. Accordingly, the source and sink terms are given by

$$\Psi_{n_c} = \frac{\partial n_c}{\partial t}\bigg|_{\text{act}} + \frac{\partial n_c}{\partial t}\bigg|_{\text{evap}} + \frac{\partial n_c}{\partial t}\bigg|_{\text{auto}} + \frac{\partial n_c}{\partial t}\bigg|_{\text{accr}} + \frac{\partial n_c}{\partial t}\bigg|_{\text{sed,c}}, \tag{54}$$

$$\Psi_{q_c} = \frac{\partial q_c}{\partial t}\bigg|_{\substack{\text{cond}\\\text{evap}}} + \frac{\partial q_c}{\partial t}\bigg|_{\text{evap}} + \frac{\partial q_c}{\partial t}\bigg|_{\text{auto}} + \frac{\partial q_c}{\partial t}\bigg|_{\text{accr}} + \frac{\partial q_c}{\partial t}\bigg|_{\text{sed,c}}. \tag{55}$$

In the following, the source/sink terms for activation, condensation and evaporation are described. This improved microphysics

was recently applied by Schwenkel and Maronga (2019) for studying nocturnal radiation fog. Besides this physical improvement, the bulk microphysics are now fully modularized in PALM 6.0.

### 3.1.1 Activation of cloud droplets

As activation is the major source term for $n_c$, this process is represented by so called Twomey-type parameterizations in PALM 6.0, which are available in two modes. Per default the number of activated cloud condensation nuclei (CCN) is given by a

simple power-law expression

$$N_{\text{CCN}} = N_a s_l^{k_{\text{act}}}, \tag{56}$$

where $N_{\text{CCN}}$ is the number of activated CCN, $N_a$ is the number concentration of the dry aerosol and the exponent $k_{\text{act}}$ depending on the type of analyzed aerosol (Twomey, 1959). The supersaturation over a liquid phase surface is given by $s_l = q_v/q_{v,\text{sat}} - 1$, where $q_{v,\text{sat}}$ stands for the water vapor saturation mixing ratio. Moreover, a more advanced method considering

physio-chemical properties of the dry aerosol can be used after Khvorostyanov and Curry (2006). Therein it is assumed that the dry aerosol spectrum follows a log-normal distribution which is given by

$$f_d = \frac{N_a}{\sqrt{2\pi}\ln\sigma_d r_d}\exp\left[-\frac{\ln^2(r_d/r_{d,\text{av}})}{2\ln^2\sigma_d}\right], \tag{57}$$

where $r_d$ and $r_{d,\text{av}}$ are the radius and the mean radius of the dry aerosol, respectively. The dispersion of the dry aerosol spectrum is displayed by $\sigma_d$. Hence, the number of activated aerosol is calculated by

$$N_{\text{CCN}}(s_l) = \frac{N_a}{2}[1 - \text{erf}(u)]; \qquad u = \frac{\ln(s_0/s_l)}{\sqrt{2}\ln\sigma_{s,l}}, \tag{58}$$

where erf is the Gaussian error function, and

$$s_0 = r_{d,\text{av}}^{-(1+\beta)}\left(\frac{4A^3}{27b}\right)^{1/2}, \qquad \sigma_s = \sigma_d^{1+\beta}. \tag{59}$$

$A$ is the Kelvin parameter and $b$ and $\beta$ depend on the chemical composition and physical properties of the soluble part of the dry aerosol. Both schemes have in common that $N_a$ must be prescribed and $n_c$ is calculated as a function of the aerosol

concentration and the supersaturation. However, for the latter scheme the physio-chemical properties, such as the mean dry radius, chemical composition and dispersion of the aerosol spectrum of the aerosol, must be prescribed by the user. The activation rate is then given by

$$\frac{\partial n_c}{\partial t}\bigg|_{\text{act}} = \max\left(\frac{N_{\text{CCN}} - n_c}{\Delta t}, 0\right), \tag{60}$$



where $n_c$ is the number of previously activated aerosols that are assumed to be equal to the number of pre-existing droplets and $\Delta t$ is the length of the model time step. However, it must be mentioned that in regions with significant autoconversion and accretion growth the subsequent depletion of $n_c$ might lead to an overprediction of activation with this method.

### 3.2 Improved representation of diffusional growth

5 Additionally, for treating condensational growth a second method (diagnostic approach) apart from the well-established saturation adjustment scheme was implemented (see Maronga et al., 2015). This method diagnoses the current supersaturation from the fields of $T$ and $q_v$. Subsequently, the diagnosed supersaturation is used for calculating the condensation and evaporation rates for cloud droplets, which is given by (Khairoutdinov and Kogan, 2000):

$$\left. \frac{\partial q_c}{\partial t}\right|_{\substack{\mathrm{cond}\\\mathrm{evap}}} = \frac{4\pi\Gamma(T,p)\rho_w n_c}{\rho_a} s_l r_c. \tag{61}$$

10 Here, $r_c$ is the volume mean radius of cloud droplets and $\Gamma$ is a function of temperature and pressure including the thermal conduction and diffusion of water vapor in air. Ventilation effects which can affect the effective evaporation rates are considered for rain droplets separately, as described in Maronga et al. (2015). Note that this diagnostic scheme is an appropriate alternative, particularly if the assumptions made for saturation adjustment (assuming equilibrium) are violated, i.e., for time steps shorter than a few seconds.

### 15 3.3 Lagrangian particle model improvements

In the last years, the embedded LPM has been successfully used to study scalar dispersion in urban environments (e.g. Auvinen et al., 2017; Lo and Ngan, 2017; Gronemeier and Sühring, 2019). The LPM is based on Weil et al. (2004) to separate the particle speed into a deterministic and a stochastic contribution, which corresponds to dividing the turbulent flow field into a resolved-scale and a SGS portion, respectively. The resolved-scale velocity is provided by the LES at each time step, while the SGS 20 velocity is predicted by integrating a stochastic differential equation according to Weil et al. (2004). For details on the model and its implementation we refer to Steinfeld et al. (2008) and Maronga et al. (2015).

As particle boundary conditions at solid walls, PALM 6.0 offers absorption and reflection boundary conditions. The particle reflection boundary conditions were revised and adjusted to the revised topography implementation where also overhanging structures may appear. Now, within a timestep, particles can be reflected multiple times at different solid walls, which is 25 especially important near building corners.

Furthermore, the LPM was adjusted to the self-nesting (see Sect. 4.8). Particles that enter the region of one of the child domains are automatically transferred from the parent to the respective child model. Vice versa, particles leaving a child domain are automatically transferred back to its parent model. A technical description of this approach as well as implications concerning the treatment of SGS particle velocities when particles are transferred between parent and child will be discussed 30 in a follow-up study.



## 3.4 Lagrangian cloud model improvements

PALM's Lagrangian cloud model (LCM) is based on its LPM, using Lagrangian particles as so-called superdroplets (e.g., Shima et al., 2009), each representing an ensemble of identical droplets that change their properties (e.g., water mass, aerosol mass, number of represented real droplets - the so-called weighting factor) by undergoing cloud microphysical processes.

PALM's approach has been applied in various studies to further process-level understanding of warm-phase cloud microphysics, covering deliquescent aerosols, their entrainment and mixing with the cloud, as well as droplet activation, growth by diffusion, and collision and coalescence (Riechelmann et al., 2012; Hoffmann et al., 2015, 2017; Hoffmann, 2017; Noh et al., 2018).

### 3.4.1 Collision and coalescence

While the modeling of aerosol activation and diffusional growth of cloud droplets is based on first principles and is very similar in all available LCMs (Andrejczuk et al., 2008; Shima et al., 2009; Riechelmann et al., 2012), the representation of collision and coalescence (i.e., collection) depends heavily on model formulation. In a recent review paper, Unterstrasser et al. (2016) compared all available representations of collection in LCMs to analytical and other benchmark solutions. They showed that PALM's previous representation of collection is very stable but significantly underestimates the growth of the largest droplets,

with commensurate effects on the initiation of rain. Therefore, our previous default collection algorithm by Riechelmann et al. (2012) was replaced by the so-called *all-or-nothing* algorithm that is based on the ideas of Shima et al. (2009) and Sölch and Kärcher (2010), and performed best in the comparison by Unterstrasser et al. (2016). The basic ideas of the all-or-nothing algorithm will be summarized below, but the interested reader is referred to Hoffmann et al. (2017) for more details on its implementation in PALM.

In the all-or-nothing approach, each real droplet of the superdroplet with the smaller weighting factor collects one real droplet of the superdroplet with the larger weighting factor. The probability $P$ of this interaction is given by

$$P_{mn} = KK_{mn}\frac{\Delta t}{\Delta V} \cdot A_{\mathrm{w},n}, \tag{62}$$

where $m$ and $n$ are the indices of the superdroplets with the smaller and larger weighting factor, respectively, $KK$ is the collection kernel depending on the properties of both superdroplets, $\Delta V$ is a prescribed volume in which the superdroplets

are allowed the collide (which equals the size of an LES grid box in PALM), and $A_{\mathrm{w}}$ is the superdroplet weighting factor. If $P_{mn}$ exceeds a random number chosen uniformly from the interval $[0, 1]$, the collection takes place. First the mass of each real droplet of superdroplet $m$ increases, while the mass of each real droplet of superdroplet $n$ remains unchanged:

$$\widehat{m}_m = m_m + m_n \qquad \text{and} \qquad \widehat{m}_n = m_n, \tag{63}$$

where the $\widehat{(..)}$ marks the variable after collection. Second, the aerosol mass of each real droplet changes:

$\widehat{m}_{s,m} = m_{s,m} + m_{s,n} \qquad \text{and} \qquad \widehat{m}_{s,n} = m_{s,n}.$ \hfill (64)





Finally, the change in the weighting factor diverges from this pattern:

$$\widehat{A}_m = A_m \qquad \text{and} \qquad \widehat{A}_n = A_n - A_m. \qquad\qquad (65)$$

This procedure is repeated for all different (unordered) superdroplet pairs in the volume $\Delta V$.

### 3.4.2 Splitting and merging of superdroplets

In recent studies with the LCM it was observed that droplet size distributions does not converge even for large numbers (approximately 200) of superdroplets per grid box (e.g., Riechelmann et al., 2012). Based on the original idea of Unterstrasser and Sölch (2014), a splitting and merging algorithm for superdroplets has been adapted for our LCM. The main goal of such algorithms is to improve statistics by splitting one superdroplet into several superdroplets with the commensurate reduction of the weighting factor, and to save computational demand by merging several superdroplets into one superdroplet if appropriate.

For a correct representation of the initiation of rain in warm clouds, a good statistical representation of collecting droplets by a sufficiently high number of superdroplets is indispensable.

The splitting algorithm is mainly steered by three parameters: 1. The minimum radius of superdroplets that will be split potentially, 2. a threshold for the weighting factor of that superdroplet (can either be prescribed or is approximated by assuming a gamma distribution, see Schwenkel et al. (2018)), and 3. the splitting factor, which describes in how many particles one

superdroplet will be split (prescribed or calculated by the LCM). However, the general splitting procedure is simple: If one superdroplet fulfills all criteria, the superdroplet is $\eta_{\mathrm{spl}} - 1$ times cloned and the weighting factor of the original and all new superdroplets is reduced to $A_{n,\mathrm{new}} = A_n / \eta_{\mathrm{spl}}$, while $\eta_{\mathrm{spl}}$ is the splitting factor, determining how many new superdroplets will be created during one operation. All other properties of the affected superdroplets remain unaffected. However, after a few timesteps every cloned superdroplet will experience slightly different subgrid scale velocities and collisional growth rates due

to the stochastic nature of these routines. Note that the splitting procedure is only applied in grid boxes where a threshold for the number of superdroplets per grid box is not exceeded to ensure computational feasibility. The merging algorithm is designed to save computational costs by merging superdroplets in regions where an increased superdroplet resolution is not required, e.g., outside of clouds. If a superdroplet grows smaller than a prescribed radius and exhibits a large enough (larger than a prescribed value) weighting factor, the superdroplet will be merged with another superdroplet in the same grid box that

also fulfills these requirements. By doing so, the first superdroplet is deleted and the weighting factor of the other superdroplet is adapted to obey mass conservation. The splitting/merging algorithm is described in detail in Schwenkel et al. (2018). Their results show that the merging algorithm improves the representation of the collection process significantly, while decreasing computational time by up to 18% compared to a simulation with a globally increased superdroplet number.

### 3.5 Land surface model (LSM)

Traditionally, LES models are used with prescribed surface conditions (either by prescribing surface fluxes or by explicitly setting surface temperature and humidity). However, in many cases, an LSM is required in which the surface fluxes have to be calculated based on the state of the solid material (soil, water, pavement), the radiation budget of the surface, and atmospheric





**Table 5.** List of symbols related to clouds and precipitation.

| Symbol | Dimension | Description |
| --- | --- | --- |
| $A$ | m | Kelvin curvature parameter |
| $A_{\mathrm{w}}$ | | Superdroplet weighting factor |
| $b$ | | Parameter describing the physiochemical properties of dry aerosol |
| $k_{\mathrm{act}}$ | | Exponent for aerosol activation with power-law expression |
| $KK$ | | Collection kernel in LCM |
| $N_{\mathrm{CCN}}$ | $\mathrm{m}^{-3}$ | Number of activated cloud condensation nuclei |
| $N_{\mathrm{a}}$ | $\mathrm{m}^{-3}$ | Number concentration of dry aerosol |
| $n_{\mathrm{c}}$ | $\mathrm{m}^{-3}$ | Cloud droplet number concentration |
| $n_{\mathrm{r}}$ | $\mathrm{m}^{-3}$ | Rain droplet number concentration |
| $P$ | | Collection probability in LCM |
| $q_{\mathrm{c}}$ | $\mathrm{kg\,kg}^{-1}$ | Cloud water mixing ratio |
| $q_{\mathrm{r}}$ | $\mathrm{kg\,kg}^{-1}$ | Rain water mixing ratio |
| $q_{\mathrm{l}}$ | $\mathrm{kg\,kg}^{-1}$ | Liquid water mixing ratio |
| $q_{\mathrm{v}}$ | $\mathrm{kg\,kg}^{-1}$ | Water vapor mixing ratio |
| $q_{\mathrm{v,sat}}$ | $\mathrm{kg\,kg}^{-1}$ | Water vapor mixing ratio at saturation |
| $r_{\mathrm{c}}$ | m | Volume mean cloud droplet radius |
| $r_{\mathrm{d}}$ | m | Droplet radius |
| $r_{\mathrm{d,av}}$ | m | Mean droplet radius |
| $s_0$ | | Supersaturation considering solute and curvature effects of the dry aerosol |
| $s_{\mathrm{l}}$ | | Supersaturation over a flat water surface |
| $\beta$ | | Parameter of the soluble fraction of the dry aerosol |
| $\Gamma$ | | Function used in cloud microphysics parameterization |
| $\eta_{\mathrm{spl}}$ | | Splitting factor in LCM |
| $\sigma_{\mathrm{d}}$ | | Standard deviation of the dry aerosol spectrum |
| $\sigma_{\mathrm{l,s}}$ | | Standard deviation of supersturation considering solute and curvature effects of the dry aerosol spectrum |
| $\Psi_{n_{\mathrm{c}}}$ | $\mathrm{kg\,kg}^{-1}\,\mathrm{s}^{-1}$ | Source/sink term of $n_{\mathrm{c}}$ |
| $\Psi_{q_{\mathrm{c}}}$ | $\mathrm{kg\,kg}^{-1}\,\mathrm{s}^{-1}$ | Source/sink term of $q_{\mathrm{c}}$ |
| $\Psi_{q_{\mathrm{v}}}$ | $\mathrm{kg\,kg}^{-1}\,\mathrm{s}^{-1}$ | Source/sink term of $q_{\mathrm{v}}$ |
| $\Psi_{s}$ | $\mathrm{kg\,m}^{-3}\,\mathrm{s}^{-1}$ | Source/sink term of $s$ |

conditions. This might be the case when respective measurement data is absent, or when the interaction between atmosphere and surface becomes relevant, e.g., in case of cloud or fog formation (Maronga and Reuder, 2017). Furthermore, LSMs are needed when the model is to be run in a forecasting mode, where surface boundary conditions are *a priori* unknown.





The implemented LSM in PALM is similar to the Tiled ECMWF Scheme for Surface Exchanges over Land (TES-SEL/HTESSEL Balsamo et al., 2009) and the derivative (simplified) implementation in the LES model DALES (Heus et al., 2010). The scheme implemented in PALM 6.0 was adapted for use with impervious surfaces (e.g. streets, pavements) as well as water surfaces, and was coupled to a radiation model (see Sect. 3.6) and both bulk cloud physics and Lagrangian cloud model (see Sect. 3.1 and Maronga et al., 2015).

The LSM consists of a solver for the energy balance of the Earth's surface using a resistance parametrization for the surface fluxes and a multi-layer soil scheme. The energy balance of the Earth's surface is calculated as

$$\frac{dT_0}{dt} C_0 = R_{\mathrm{n}} - H - LE - G \tag{66}$$

where $C_0$ and $T_0$ are the heat capacity and radiative temperature of the surface skin layer, respectively. Note that $C_0$ is usually zero as it is assumed that the skin layer does not have a heat capacity (see below). $R_{\mathrm{n}}$, $H$, $LE$, and $G$ are the net radiation, sensible heat flux, latent heat flux, and ground (soil) heat flux at the surface, respectively.

$H$ is calculated as

$$H = -\rho c_p \frac{1}{r_{\mathrm{a}}} (\theta_{\mathrm{mo}} - \theta_0) \tag{67}$$

where $r_{\mathrm{a}}$ is the aerodynamic resistance. $\theta_0$ and $\theta_{\mathrm{mo}}$ are the potential temperature at the surface and at a fixed height within the atmospheric surface layer (at height $z_{\mathrm{mo}}$) respectively. $r_a$ is calculated via MOST as

$$r_{\mathrm{a}} = \frac{\theta_{\mathrm{mo}} - \theta_0}{u_* \, \theta_*} . \tag{68}$$

$G$ is parameterized as (Duynkerke, 1999)

$$G = \Lambda (T_0 - T_{\mathrm{soil},1}), \tag{69}$$

with $\Lambda$ being the total thermal conductivity between skin layer and the uppermost soil layer. $T_0$ is the radiative surface temperature (related to the radiative potential temperature via the Exner function) and $T_{\mathrm{soil},1}$ is the temperature of the uppermost soil layer (calculated at the center of the layer). $\Lambda$ is calculated via a resistance approach as a combination of the conductivity between the canopy and the soil-top (constant value) and the conductivity of the top half of the uppermost soil layer:

$$\Lambda = \frac{\Lambda_{\mathrm{skin}} \Lambda_{\mathrm{soil}}}{\Lambda_{\mathrm{skin}} + \Lambda_{\mathrm{soil}}} . \tag{70}$$

When no skin layer is used (i.e. in case of bare soil and pavements), $\Lambda$ reduces to the heat conductivity of the uppermost soil layer (divided by the layer depth). In that case, it is assumed that the soil temperature is constant within the uppermost 25 % of the top soil layer and equals the radiative temperature at the surface. $C_0$ is then set to a non-zero value according to the material properties. The latent heat flux $LE$ is calculated as

$$LE = -\rho \, l_{\mathrm{v}} \, \frac{1}{r_{\mathrm{a}} + r_{\mathrm{s}}} (q_{\mathrm{v,mo}} - q_{\mathrm{v,sat}}(T_0)) . \tag{71}$$





Here, $r_\text{s}$ is the surface resistance, $q_{v,mo}$ is the water vapor mixing ratio at height $z_\text{mo}$, and $q_{v,\text{sat}}$ is the water vapor mixing ratio at saturation.

All equations above are solved locally for each surface element of the model grid. Each element for the surface type vegetation can consist of patches of bare soil, vegetation, and a liquid water reservoir, which is the interception water stored on plants

from precipitation. Therefore, an additional equation is solved for the liquid water reservoir. A liquid water reservoir is also available when the surface type is set to pavement, representing the ability of impervious surfaces to store a limited amount of liquid water at the surface. $LE$ is then calculated for each of the three components (bare soil, vegetation, liquid water on plants/pavements). The resistances are calculated separately for bare soil and vegetation following Jarvis (1976).

For water surfaces, PALM currently only allows for prescribing a bulk water temperature. The energy balance is then solved

as for land surfaces, but without evapotranspiration from vegetation and bare soil. A skin layer is adopted so that $C_0 = 0$ and $\Lambda = 1 \cdot 10^{11}$ in order to calculate a reasonable heat flux into the water body.

The surface is coupled to a 1-D soil model which is called independently for each surface element. By default, the soil model consists of eight layers with default layer depths of 0.01, 0.02, 0.04, 0.06, 0.14, 0.26, 0.54, and 1.86 m, (a variable number of layers and depths can be prescribed by the user) and in which the vertical heat and water transport is modelled using the Fourier

law of diffusion and Richards' equation, respectively. Hydraulic conductivities are calculated after van Genuchten (1980). For vegetated surface elements, root fractions can be assigned to each soil layer to account for the explicit water withdrawal of plants used for transpiration from the respective soil layer. Viterbo and Beljaars (1995) and Balsamo et al. (2009) give more details.

Pavements are treated as a common soil (allowing varying depths of the pavement layers) but with physical properties of the

pavement material. The pavement layer is impermeable and prohibits the vertical transport of soil moisture.

A first validation of a previous version of the LSM for simulation of nocturnal radiation fog is given in Maronga and Reuder (2017). A detailed description of the current LSM and a validation for full diurnal cycle LES, see Gehrke et al. (2019) (to be submitted to this special issue).

### 3.6   Radiation model

In simulations with LSM, or in cases where radiative effects of clouds are of interest, a suitable radiation parameterization is essential. This involves primarily the calculation of the surface radiation budget, but also all radiative effects of clouds. PALM offers a built-in simple and fast radiation model for clear sky conditions that neglects the presence of humidity, clouds, and variations in aerosol and trace gas properties in the atmosphere. Moreover, PALM provides an interface to the shortwave and longwave components of the Rapid Radiative Transfer Model (for Global models) (RRTMG; e.g. Clough et al., 2005). Both

options calculate the radiation budget of the Earth's surface, which reads

$$R_\text{n} = SW_\downarrow - SW_\uparrow + LW_\downarrow - LW_\uparrow \tag{72}$$

where $R_\text{n}$ is net radiation. $SW_\downarrow$, $SW_\uparrow$, $LW_\downarrow$, $LW_\uparrow$ are the shortwave incoming (downward), shortwave outgoing (upward), longwave incoming (downward), and longwave outgoing (upward) fluxes, respectively.




### 3.6.1 Clear-sky radiation model

The clear-sky radiation model is a simple parameterization and limited to the calculation of the radiation budget at the surface. We recommend to use this scheme only for cases in which clouds are absent and in which direct cooling or heating of air due to divergence of the radiative fluxes is negligible. In the clear-sky model, $SW_\downarrow$ is calculated based on the position of the sun and orbital parameters:

$$SW_\downarrow = S_0 \, \tau \, \sin(\Psi) \tag{73}$$

with $S_0 = 1368 \text{ W m}^{-2}$ and $\Psi$ being the solar constant and the cosine of the solar zenith angle, respectively. The transmissivity of the atmosphere $\tau$ is estimated to be

$$\tau = 0.6 + 0.2 \, \sin(\Psi) \, . \tag{74}$$

$\Psi$ depends on UTC time $t_{\text{UTC}} \in [0, 86400]$ (in seconds), day of the year $doy \in [1, 365]$ and location, defined by geographical latitude $\phi \in [-90^\circ, 90^\circ]$ and longitude $\lambda \in [-180^\circ, 180^\circ]$. $\Psi$ is calculated via

$$\Psi = \sin(\phi)\sin(d) + \cos(\phi)\cos(d)\cos(h) \, , \tag{75}$$

with

$$d = \arcsin\left[d1 \cdot \sin\left(d2 \cdot doy - d3\right)\right] \tag{76}$$

where $d$ is the declination of the sun, with

$$d1 = \sin\left(\frac{23.45 \cdot \pi}{180}\right) \, , \quad d2 = \frac{2\pi}{365} \, , \quad d3 = 81 \cdot d2 \, ; \tag{77}$$

and the hour angle being given by

$$h = 2\pi \left(\frac{t_{\text{UTC}}}{86400}\right) + \lambda - \pi \, . \tag{78}$$

The flux $SW_\uparrow$ depends on the incoming radiation $SW_\downarrow$ and surface broadband albedo $\alpha_{\text{bb}}$ (see Sect.3.6.3):

$$SW_\uparrow = \alpha_{\text{bb}} \, SW_\downarrow \, . \tag{79}$$

The flux $SW_\uparrow$ is calculated from the Stefan–Boltzmann law:

$$LW_\uparrow = \epsilon_0 \, \sigma_{\text{SB}} \, T_0^4 \tag{80}$$

where $\epsilon_0$ is the surface emissivity and $\sigma_{\text{SB}} = 5.67 \cdot 10^{-8} \text{ W m}^{-2} \text{ K}^{-4}$ is the Stefan-Boltzmann constant.

The longwave incoming radiative flux is parameterized by a first-guess approximation:

$$LW_\downarrow = \epsilon_{\text{atm}} \, \sigma \, T(z_{\text{mo}})^4 \tag{81}$$

with $\epsilon_{\text{atm}} = 0.8$ being the bulk emissivity of the atmosphere.





### 3.6.2 Coupling to RRTMG

As an advanced alternative to the clear-sky model, PALM can be used in combination with the RRTMG radiation code. The RRTMG source code is shipped along with PALM, but it is not part of the model (meaning that RRTMG is put under its own licence). Unlike most embedded modules in PALM, the RRTMG is thus used as external library and linked to the default PALM

code. The RRTMG subroutines are called from PALM as a 1-D model for each vertical column of the model grid. Vertical profiles of pressure, temperature, and water vapor volume mixing ratio are calculated in PALM and transferred to the radiation model. Moreover, information on clouds, namely the liquid water path and the effective droplet radius, are calculated by PALM and transferred to RRTMG. Concentration of other trace gases are read during intialization from an input file. The standard profiles shipped with RRTMG are used by default. As RRTMG requires data up to the top of the atmosphere, the PALM data

(i.e., $\theta, p$, etc.) is extended and interpolated in the vertical direction by standard profiles, e.g., those shipped with RRTMG. In order to avoid large gradients at the interface between PALM's upper boundary and the upper atmospheric profiles, profiles are gradually blended over within the five grid points above the PALM domain.

RRTMG provides the shortwave and longwave radiative heating rates for each grid volume and the surface energy budget terms (see Sect. 3.5) and was first applied coupled to PALM by Maronga and Reuder (2017). When topographical elements are

present at the surface, e.g., hills or buildings in urban area, it provides the radiation fluxes at the top boundary of the model for the radiative transfer model (RTM), see Sect. 4.4. As the default implementation of RRTMG does not provide the direct and diffuse irradiance, which are required for RTM, the original source code was modified so that they were available as outputs.

### 3.6.3 Calculation of surface albedos

The calculation of the surface albedo components for longwave diffuse $\alpha_{\mathrm{lw,dif}}$, longwave direct $\alpha_{\mathrm{lw,dir}}$, shortwave diffuse

$\alpha_{\mathrm{sw,dif}}$, and shortwave direct $\alpha_{\mathrm{sw,dir}}$ radiation, as required by RRTMG, are parameterized according to Briegleb (1986) and Briegleb (1992). The parameterization involves dynamically changing the direct radiation albedos depending on $\Psi$ while diffuse radiation albedos are taken from a look-up table (see Tab. 7). The particular calculation of the albedo depends on a surface type classification into classes ocean, sea ice, snow, asphalt, and land surfaces with strong as well as weak zenith dependence of the albedo.

For ocean surface, the the direct radiation albedo is calculated as

$$\alpha_{\mathrm{lw,dir}} = \alpha_{\mathrm{sw,dir}} = \frac{0.026}{\Psi^{1.7} + 0.065} + 0.15(\Psi - 0.1)(\Psi - 0.5)(\Psi - 1) \tag{82}$$

Snow surfaces have a zenith dependence, viz.

$$\alpha_{\mathrm{lw,dir}} = \begin{cases} 0.5(1 - \alpha_{\mathrm{lw,dif}})\dfrac{3}{1 + 4\Psi} - 1 & \text{for } \Psi < 0.5, \\ \alpha_{\mathrm{lw,dif}} & \text{for } \Psi \geq 0.5, \end{cases} \tag{83}$$





and

$$\alpha_{\mathrm{sw,dir}} = \begin{cases} 0.5(1-\alpha_{\mathrm{sw,dif}})\dfrac{3}{1+4\Psi} - 1 & \text{for } \Psi < 0.5, \\[2ex] \alpha_{\mathrm{sw,dif}} & \text{for } \Psi \geq 0.5, \end{cases} \tag{84}$$

but have additionally an upper bound limit of $0.98$. Albedos for land surface types are calculated as

$$\alpha_{\mathrm{lw,dir}} = \begin{cases} \dfrac{\alpha_{\mathrm{lw,dif}} \cdot 1.4}{1+0.8\Psi} & \text{for strong zenith dependence,} \\[2ex] \dfrac{\alpha_{\mathrm{lw,dif}} \cdot 1.1}{1+0.2\Psi} & \text{for weak zenith dependence,} \end{cases} \tag{85}$$

and

$$\alpha_{\mathrm{sw,dir}} = \begin{cases} \dfrac{\alpha_{\mathrm{sw,dif}} \cdot 1.4}{1+0.8\Psi} & \text{for strong zenith dependence.} \\[2ex] \dfrac{\alpha_{\mathrm{sw,dif}} \cdot 1.1}{1+0.2\Psi} & \text{for weak zenith dependence.} \end{cases} \tag{86}$$

Direct radiation albedos for surface of type sea ice, asphalt, and bare soil are set to be equal to those specified for diffuse radiation.

### 3.7 Wind turbine model

The rapid development of wind energy in the last two decades and the clear trend towards larger wind turbines and larger wind farms led to an increased interest in wake effects. The wake of a wind turbine is the region downstream of a wind turbine which is mainly characterised by a reduced wind speed and increased turbulence compared to the free stream. This means that a downstream wind turbine will produce less power if located in the wake of an upstream turbine. Increased wind shear and turbulence implies higher loads for downstream turbines, which can reduce the lifespan of turbine components. Wake effects are especially crucial in large wind farms where wakes of multiple turbines overlap. A detailed understanding of the wake effect and how the ABL affects the wake and vice versa is therefore very important for the wind industry from turbine manufacturers to planning companies to wind farm operators and traders. As turbulent processes in the ABL greatly affect wind turbine wakes (Vollmer et al., 2016), it may seem obvious to investigate wind farm flows with LES which can explicitly resolve most of the wake turbulence and its interaction with the ABL turbulence and thus serve as a virtual laboratory.

The wind turbine model (WTM) included in PALM is based on the common actuator disk model (ADM) approach in which the rotor of a wind turbine is represented by a permeable disk that extracts energy from the flow by applying a thrust force at the disk. While in the frequently used simple version of the ADM (e.g. as proposed by Calaf et al. (2010)) the forces are uniformly distributed and only the thrust force is considered (thus ignoring the torque), the WTM provides an advanced ADM (ADM-R) that considers varying forces over the rotor disk and rotation of the rotor blades, although these are not resolved. The basic concept is similar to the ADM-R proposed by Wu and Porté-Agel (2011) with several modifications. The rotor plane is divided into rings and segments (see Figure 1). The segments have an equal size which is a function of the grid spacing. For each segment the local lift and drag forces $f_{\mathrm{l}}$ and $f_{\mathrm{d}}$ per unit area are calculated:

$$f_{\mathrm{l}} = \frac{1}{2}\rho U_{\mathrm{rel}}^2 c_{\mathrm{l}} \frac{N_{\mathrm{b}}c}{2\pi r_{\mathrm{seg}}} \;,\;\; f_{\mathrm{d}} = \frac{1}{2}\rho U_{\mathrm{rel}}^2 c_{\mathrm{d}} \frac{N_{\mathrm{b}}c}{2\pi r_{\mathrm{seg}}}. \tag{87}$$





$U_{\mathrm{rel}}$ is the local relative velocity in the center of the segment. It is calculated from the local wind speed components (interpolation from the nearest grid points) and the velocity of the rotor blades. $c_{\mathrm{l}}$ and $c_{\mathrm{d}}$ are the lift and drag coefficients of the blades, respectively, which vary along the blade. The factor $N_{\mathrm{b}}c/(2\pi r_{\mathrm{seg}})$ represents the solidity ratio of the rotor blades (with the number of rotor blades $N_{\mathrm{b}}$, the chord $c$ and the distance of the segment from the center of the rotor disk $r_{\mathrm{seg}}$). The blade

properties have to be specified by the user. By default, the WTM includes publicly available data for the NREL 5 MW reference turbine (Jonkman et al., 2009). In a second step, the lift and drag forces are projected onto the axial and tangential planes, multiplied by a factor of -1 (Newton's third law) and then smeared and interpolated to the PALM grid points. To optimise the performance of the time-consuming smearing process, the smearing is done with a polynomial function instead of the standard Gaussian smearing and is confined to the region around the rotor. The effect of the tower and nacelle are considered by a simple

drag force approach:

$$f_{\mathrm{d,t}} = -\frac{1}{2}\rho u_{\mathrm{h}}^2 c_{\mathrm{d,t}} \ , \ \ f_{\mathrm{d,n}} = -\frac{1}{2}\rho u_{\mathrm{h}}^2 c_{\mathrm{d,n}} \tag{88}$$

where $f_{\mathrm{d,t}}$ and $f_{\mathrm{d,n}}$ are the drag forces of the tower and the nacelle, respectively, with the drag coefficients $c_{\mathrm{d,t}}$ and $c_{\mathrm{d,n}}$.

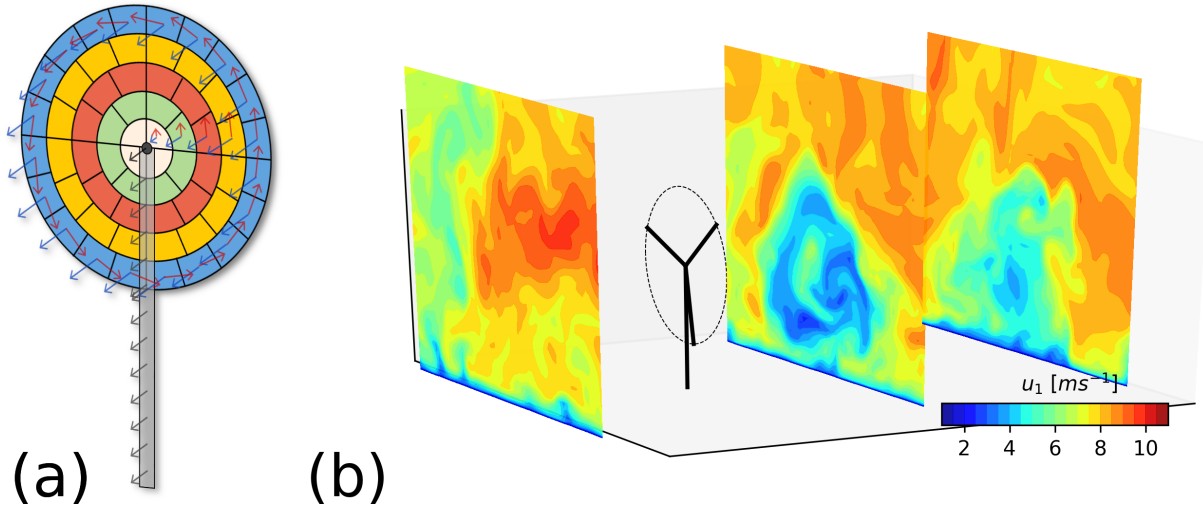

**Figure 1.** (a) schematic representation of the ADM-R model used in the WTM. Arrows denote the direction of the forces acting on the flow. (b) example of a wake simulated with PALM-WTM. Shown is the instantaneous wind speed in mean flow direction ($u$-component, inflow from the left).

The WTM contains a baseline rotational speed controller for the rotor of the wind turbine, implemented after Jonkman et al. (2009). The controller parameters are only valid for the NREL 5 MW reference turbine and need to be adjusted for different

turbine types. The controller ensures that the wind turbine's power is following the constructor's power curve by adjusting the generator torque below and the blade pitch angles above the rated wind speed of the wind turbine. The wind turbine's electrical power is calculated from the torque and rotational speed of the rotor with the possibility to correct for generator and gearbox





efficiency. A yaw controller that ensures the automatic orientation of the wind turbine perpendicular to the wind is implemented following Storey et al. (2013).

Figure 1 (right) shows a single turbine wake simulated by the PALM-WTM. The leftmost plane displays the turbulent inflow field. The near wake with its ring-shaped structure is visible in the central plane while the far wake shown in the rightmost

5   plane is more uniform and the flow is starting to recover.

The WTM has already been used in a number of studies. Dörenkämper et al. (2015) simulated the offshore wind farm EnBW Baltic 1 and investigated the impact of the stable ABL on power production and wake effects. Vollmer et al. (2016) investigated the deflected wake behind a yawed wind turbine for different atmospheric stabilities. Vollmer et al. (2017) tried to reproduce the wind conditions around an offshore wind turbine by forcing the simulation with time series from a mesoscale

10  model. They found a generally good agreement when comparing the PALM results with data from a meteorological mast and LiDAR measurements. Andersen et al. (2015) compared the results of the WTM and other LES models for very large idealised wind farms and explored how to best present and compare the resulting variability.





**Table 6.** List of symbols related to radiation and the surface energy balance.

| Symbol | Dimension | Description |
|---|---|---|
| $C_0$ | $\mathrm{J\,m^{-2}\,K^{-1}}$ | Heat capacity of the surface (skin layer) |
| $d$ | ° | Declination of the sun |
| $d_1, d_2, d_3$ | ° | Model parameters in the radiation model |
| $G$ | $\mathrm{W\,m^{-2}}$ | Soil heat flux |
| $H$ | $\mathrm{W\,m^{-2}}$ | Surface sensible heat flux |
| $h$ | ° | Hour angle |
| $LE$ | $\mathrm{W\,m^{-2}}$ | Surface latent heat flux |
| $LW_\downarrow$ | $\mathrm{W\,m^{-2}}$ | Incoming longwave radiation at the surface |
| $LW_\uparrow$ | $\mathrm{W\,m^{-2}}$ | Outgoing longwave radiation at the surface |
| $R_\mathrm{n}$ | $\mathrm{W\,m^{-2}}$ | Net radiation at the surface |
| $r_\mathrm{a}$ | $\mathrm{s\,m^{-1}}$ | Aerodynamic resistance |
| $r_\mathrm{s}$ | $\mathrm{s\,m^{-1}}$ | Surface resistance |
| $SW_\downarrow$ | $\mathrm{W\,m^{-2}}$ | Incoming shortwave radiation at the surface |
| $SW_\uparrow$ | $\mathrm{W\,m^{-2}}$ | Outgoing shortwave radiation at the surface |
| $T_{0,\mathrm{eff}}$ | K | Effective surface temperature |
| $T_{\mathrm{soil},1}$ | K | Soil temperature of the uppermost layer |
| $t_\mathrm{UTC}$ | s | UTC time in seconds since midnight |
| $\alpha_\mathrm{bb}$ | | Broadband albedo |
| $\alpha_\mathrm{eff}$ | | Effective albedo |
| $\alpha_\mathrm{lw,dif}$ | | Longwave diffuse albedo |
| $\alpha_\mathrm{lw,dir}$ | | Longwave direct albedo |
| $\alpha_\mathrm{sw,dif}$ | | Shortwave diffuse albedo |
| $\alpha_\mathrm{sw,dir}$ | | Shortwave direct albedo |
| $\epsilon_0$ | | Surface emissivity |
| $\epsilon_{0,\mathrm{eff}}$ | | Effective surface emissivity |
| $\theta_\mathrm{sp,av}$ | K | Mean potential temperature during surface spinup |
| $\theta_\mathrm{sp,amp}$ | K | Amplitude of the potential temperature sinusoidal forcing during surface spinup |
| $\Lambda$ | $\mathrm{W\,m^{-2}K^{-1}}$ | Total thermal conductivity between the surface and the uppermost soil layer |
| $\Lambda_\mathrm{skin}$ | $\mathrm{W\,m^{-2}K^{-1}}$ | Thermal conductivity of the skin layer |
| $\Lambda_\mathrm{soil}$ | $\mathrm{W\,m^{-2}K^{-1}}$ | Thermal conductivity of the uppermost soil layer |
| $\lambda$ | ° | Geographical longitude |
| $\tau$ | | Transmissivity of the atmosphere |
| $\phi$ | ° | Geographical latitute |
| $\Psi$ | | Cosine of the solar zenith angle |
| $\Xi$ | | Decoupling factor used in plant canopy transpiration |





**Table 7.** Albedos for a solar angle of $80^\circ$ for different surface types.

| Surface type | Shortwave albedo | Longwave albedo | Broadband albedo | Zenith dependence |
|---|---|---|---|---|
| ocean | 0.06 | 0.06 | 0.06 | ocean-specific |
| mixed farming, tall grassland | 0.09 | 0.28 | 0.19 | strong |
| tall/medium grassland | 0.11 | 0.33 | 0.23 | strong |
| evergreen shrubland | 0.11 | 0.33 | 0.23 | strong |
| short grassland/meadow/shrubland | 0.14 | 0.34 | 0.25 | weak |
| evergreen needleleaf forest | 0.06 | 0.22 | 0.14 | weak |
| mixed deciduous forest | 0.06 | 0.27 | 0.17 | weak |
| deciduous forest | 0.06 | 0.31 | 0.19 | weak |
| tropical evergreen broadleaved forest | 0.06 | 0.22 | 0.14 | weak |
| medium/tall grassland/woodland | 0.06 | 0.28 | 0.18 | weak |
| desert, sandy | 0.35 | 0.51 | 0.43 | strong |
| desert, rocky | 0.24 | 0.40 | 0.32 | strong |
| tundra | 0.10 | 0.27 | 0.19 | strong |
| land ice | 0.90 | 0.65 | 0.77 | weak |
| sea ice | 0.90 | 0.65 | 0.77 | none |
| snow | 0.95 | 0.70 | 0.82 | snow-specific |
| bare soil | 0.08 | 0.08 | 0.08 | none |

**Table 8.** List of symbols related to the wind turbine model.

| Symbol | Dimension | Description |
|---|---|---|
| $c$ | m | Chord length in wind turbine model |
| $c_\mathrm{d}$ | | Blade drag coefficient in wind turbine model |
| $c_\mathrm{l}$ | | Blade lift coefficient in wind turbine model |
| $c_\mathrm{d,n}$ | | Nacelle drag coefficient in wind turbine model |
| $c_\mathrm{d,t}$ | | Tower drag coefficient in wind turbine model |
| $f_\mathrm{d}$ | $\mathrm{N\,m^{-2}}$ | Blade drag force in wind turbine model |
| $f_\mathrm{l}$ | $\mathrm{N\,m^{-2}}$ | Blade lift force in wind turbine model |
| $f_\mathrm{d,n}$ | $\mathrm{N\,m^{-2}}$ | Nacelle drag force in wind turbine model |
| $f_\mathrm{d,t}$ | $\mathrm{N\,m^{-2}}$ | Tower drag force in wind turbine model |
| $N_\mathrm{b}$ | | Number of rotor blades in wind turbine model |
| $r_\mathrm{seg}$ | m | Distance of a segment from the rotor disk center in the wind turbine model |
| $U_\mathrm{rel}$ | $\mathrm{m\,s^{-1}}$ | Local relative velocity in wind turbine model |





**Table 9.** List of lateral boundary conditions available in PALM.

| boundary condition | types |
| --- | --- |
| cyclic | - |
| inflow | laminar |
|  | turbulence recycling |
|  | synthetic turbulence generator |
| outflow | radiation |
|  | turbulent outflow |



## 4 PALM-4U components

In this section those modules are described which were newly implemented in the PALM model system. Note that PALM-4U components are essentially designed to be used for applications in urban environment, although they might as well be valuable for simulations without clear focus on urban processes.

### 4.1 Topography

Cartesian topography in PALM is considered using the mask method (Briscolini and Santangelo, 1989), where a grid cell is either 100% fluid or 100% obstacle. Fluid grid points are divided into grid without adjacent surfaces where standard equations are solved, and surface-adjacent grid points where partly different code is executed, e.g., to represent wall functions. In former PALM versions the topography masking was implemented using 2-D horizontal maps of vertical cell indices indicating the topography top at each grid point $(x, y)$ on the staggered grid, effectively transforming a 3-D building and terrain topology into a 2.5-D topography. In this way, Cartesian topography was limited to surface mounted obstacles, forbidding overhanging structures such as bridges or tunnels. In order to overcome this limitation, PALM 6.0 uses a 3-D bit array to flag obstacles and surface-adjacent grid points. The individual terms of the standard governing equations are then solved at every model grid point. Obstacle grid points and, if required, surface-adjacent grid points are masked by multiplying the individual terms of the standard equations with zero. This revised Cartesian topography implementation enables full 3-D representation of obstacles, allowing for considering bridges, or tunnel-like openings within buildings as recently studied by Gronemeier and Sühring (2019). A new Fortran derived-type data structure was implemented to efficiently store and compute data for complex surfaces (see Sect. 5.1).

### 4.2 Gas phase chemistry

Gas phase chemistry has been implemented into PALM 6.0 as an optional feature. When the gas phase chemistry option is invoked, $N_{\mathrm{chem}}$ additional equations in analogy to Eq. 5 will be solved with $N_{\mathrm{chem}}$ being the number of variable compounds of the chemical reaction scheme. The source/sink term therein for each chemical species includes emissions, chemical transformation, and deposition. This implementation permits the choice amongst gas phase chemistry schemes of different complexity. Automatic generation of the chemistry code with the Kinetic Pre-Processor, KPP version 2.2.3 (Damian et al., 2002) and an adapted version of the KP[4] preprocessing tool (Jöckel et al., 2010) allows for high flexibility in the choice of gas phase chemical mechanisms. Photolysis frequencies for reactions involving photochemistry are parameterized according to Saunders et al. (2003). Details of the chemistry implementation in PALM-4U are given by Khan et al. (2019) (to be submitted to this special issue).

A number of predefined gas phase chemical mechanisms of different complexity ranging from the photo-stationary state to CBM4 (Gery et al., 1989) are supplied with PALM. The source code for the chosen gas phase chemistry mechanism is generated by utilizing the preprocessor prior to the compilation of the PALM-4U code. Due to the high computational demands, a compromise is necessary with respect to the degree of detail of the atmospheric chemistry. By default, the source code of



PALM-4U is supplied with a chemistry module describing the photo-stationary equilibrium between NO, $O_3$, and NO, plus the transport of a passive compound. The parameterization of dry deposition processes is implemented following the resistance approach. For gaseous compounds the DEPAC module (Van Zanten et al., 2010) is used. Deposition of aerosols (PM10, PM2.5) were implemented following Zhang et al. (2001).

Emissions of gases and/or passive compounds can be provided in different levels of detail (LOD). Three modes were implemented of which two require emission data from file and one is defined via a Fortran namelist and information from the static driver (see also Sect. 5.2.1). The latter, named 'Parameterized' mode, is currently only implemented for the traffic sector, in which emissions rely on a street type classification provided by OpenStreetMap[1]. For data from file gridded emission information can be provided in two LODs in the desired netCDF format file (see also Sect. 5.2.1). LOD 1 files require annual emission

information which are temporally disaggregated by PALM-4U using sector-specific time factors while the LOD 2 files must contain temporally disaggregated emission information. Details of these three emission modes are provided in the online documentation of PALM-4U chemistry module. A case study applying the chemistry implementation of PALM-4U using gridded real-time traffic emission data is presented in Banzhaf et al. (2019) (to be submitted to this special issue).

### 4.3  Aerosol physics

Aerosol physics were implemented based on the sectional aerosol module for large-scale applications (SALSA, Kokkola et al., 2008) which includes a detailed description of the aerosol number size distribution, chemical composition and aerosol dynamic processes. This very aerosol module was chosen to be implemented in PALM due to SALSA's flexibility, and particularly since one major criteria in its development has been limiting computational expenses without the cost of accuracy.

In the aerosol module, a continuous aerosol size distribution function is discretized into a number of size bins (10 by default)

based on the mean particle diameter, and each bin is further divided into different chemical components. The number and mass concentration of each chemical component in each bin are prognostic variables. Currently, the following chemical components can be included: sulphuric acid ($H_2SO_4$), organic carbon (OC), black carbon (BC), nitric acid ($HNO_3$), ammonium ($NH_3$), sea salt, dust and water ($H_2O$). By default aerosol particles in one bin are considered internally mixed but it is possible to include externally mixed populations as well. The aerosol dynamic processes included are coagulation, nucleation, dry deposition on

solid surfaces and resolved-scale vegetation, and condensation and dissolutional growth by gaseous $H_2SO_4$, $HNO_3$, $NH_3$ and semi- and non-volatile organics (SVOC and NVOC). These gas concentration can be read to SALSA from the online chemistry module (section 4.2).

The model implementation has been successfully evaluated against measurements on the vertical variation of the aerosol number size distribution and concentration in an urban environment. For details of the implementation and model evaluation,

see Kurppa et al. (2019, this special issue).

---

[1]https://www.openstreetmap.org





## 4.4 Radiative transfer in complex environments

The Radiative Transfer Model (RTM) in PALM 6.0 calculates radiative interactions in geometrically complex environments like street-level urban canopy or complex terrain, and it represents a key component in modelling of energy exchanges for such scenarios. The RTM takes radiation from the radiation model (e.g. clear-sky or RRTMG, see Sect.3.6) on top of the complex

urban or natural canopy layer and it models the shortwave and longwave radiative processes inside this layer. It resolves shading, multiple reflections, emission and absorption of radiation among surfaces and volumetric plant canopy within three-dimensional geometry. The resulting radiative fluxes are then supplied to the surface energy balance in the LSM and BSM modules. Sensible heat from radiation absorbed inside plant canopy is used to calculate the tendencies of the air potential temperature in volumes occupied with vegetation. The radiative fluxes inside the plant canopy are also used for the calculation

of evapotranspiration and corresponding latent heat from trees. Moreover, the RTM models the Mean Radiant Temperature and provides correponding SW and LW fluxes to the biometeorology module for calculation of biometeorology indices (see Sect. 4.11).

The first version of RTM 1.0 appeared in PALM as a part of PALM-USM module (see Resler et al., 2017). The current version of the RTM 3.0 which is part of PALM-4U 6.0 was significantly enhanced and it serves for calculation of radiative

exchange among all surfaces (BSM as well as LSM surfaces). The enhancements of the new version include incorporation of new processes (e.g. interaction of longwave radiation with plant canopy), improved accuracy and scalability with new angular discretization scheme as well as the significant improvement of performance via a new 2-D raytracing method and highly optimized parallelization. All these enhancements will be described in detail in Krč et al. (2019, to be submitted to this special issue).

### 4.4.1 Modelling of radiative processes

The RTM simulates radiative processes by calculating radiative fluxes between the sun, sky, individual grid surface elements and individual grid cells containing plant canopy. All radiative fluxes are modelled separately for shortwave and longwave radiation. The irradiance of each surface element is calculated using *view factors* (VF), particularly sky view factors for diffuse radiation from the sky and surface view factors for reflected and emitted radiation. The plant canopy interaction with radiation

is modelled via *plant canopy sink factors* (CSF) which represent the portion of the radiation arriving from a particular source (e.g. the sun, the sky or a surface element) which is absorbed in a particular plant canopy grid cell.

The RTM uses the computational domain of PALM and its discretization and splitting among parallel MPI processes. The calculation of radiative interactions is split into two phases. The calculation of VF and CSF and other time-invariant data, which represents a computationally demanding process, is done once during the initialization phase of the model so that the

CPU, memory and IPC demands for the following time-stepping phase are minimized. The calculation of the actual irradiances and heat fluxes is done during each radiation time step and it represents a computationally inexpensive process.





### 4.4.2 View factors and canopy sink factors

The RTM 3.0 uses raytracing together with angular discretization of view for calculation of radiative interactions. An optimized and parallelized 2-D raytracing algorithm is used to follow a fixed number of rays from each surface element or plant canopy grid cell, each of them representing an analytically determined portion of its view. The surface element that each ray strikes is

used as a partial source of the target face's irradiance. The portion of view represented by the rays targeted above the horizon forms the target face's sky view factor, which describes irradiance by diffuse solar radiation and by the thermal radiation from the sky. The direct solar irradiance and shading is solved using precomputed ray paths for discretized apparent solar positions for simulation times.

Plant canopy is resolved as a fully three-dimensional structure of grid cells, each of which can have different leaf area density

and therefore different optical properties. The partial opacity of plant canopy grid cells means that the leaves of the trees and shrubs cover a portion of the view from the respective surface elements in the direction of the grid cell. When raytracing, the grid cells with plant canopy are considered as partially opaque obstacles with transmittance determined as

$$T = \frac{\Phi_e^t}{\Phi_e^i} = e^{-\alpha as}, \tag{89}$$

where $\Phi_e^i$ is the radiant flux carried by the ray as it enters the grid cell, $\Phi_e^t$ is the radiant flux carried by the ray as it leaves the

grid cell, $a$ is the leaf area density and $s$ is the length of the ray's intersection with the grid cell and $\alpha$ is a constant extinction coefficient, which converts LAD of trees and shrubs into a corresponding average optical density. This information, determined during raytracing, is stored as plant canopy sink factors. These factors are used to calculate the heat flux absorbed by plant canopy and also the longwave radiative flux emitted by the plant canopy in the direction of the respective surface element.

The process of each raytracing represents a challenge for distributed memory parallel processing as it requires data from

different parallel subdomains stored in different MPI processes. With a fixed amount of traced rays per surface element in angular discretization, the amount of view factors grows with $\mathcal{O}(n^2)$ and the amount of canopy sink factors grows with $\mathcal{O}(n^3)$ if the resolution of the domain is increased by the factor of $n$ in each dimension. Therefore, the computational and memory complexity of both the raytracing process and the time-stepping part of RTM is in pair with scaling of other PALM-4U processes and it represents a significant improvement over RTM 1.0. The optimized 2-D raytracing algorithm processes all

rays directed in one azimuth at once, which helps to significantly decrease computational demands and to optimize MPI data exchange patterns. With these optimizations, the time spent in the RTM 3.0 represents a marginal portion of the total PALM-4U simulation time for typical scenarios (i.e., less than 5 % of the total computing time for the largest case tested withdomain with an horizontalextent of $1800 \times 1800$ m and 2 m resolution covering complex terrain of Prague center running at 1296 CPU cores).All optimizations and their effects are discussed in detail in a separate paper dedicated to RTM 3.0.

### 4.4.3 Irradiances and absorption of radiation by plant canopy

The calculation of the irradiances and radiative heat fluxes is done during each radiation time step by the application of the precomputed view factors and canopy sink factors. The process starts by calculation of direct and diffuse radiation from the sun





and the sky and it continues with the calculation of the configured number of the reflections after which all remaining radiative flux is considered as absorbed. The remaining flux can be verified in model outputs to be negligible.

Plant leaves have very high surface-to-mass ratio and they readily exchange heat with surrounding air via convection and evapotranspiration. Their temperature hence usually differs only marginally from the surrounding air. The current implementa-
tion of the RTM considers leaves as having zero thermal capacity and identical temperature as the surrounding air, which means that the difference of heat flux from absorbed minus emitted radiation is directly transferred to the air mass. This simplification represents a common approach (see e.g. Dai et al., 2003).

Figure 2 illustrates the respective components of longwave and shortwave irradiance for a typical urban scenario during the summer day. It can be seen that the diffuse longwave irradiance from the sky and the thermal irradiance from other surfaces and
plant canopy complement each other and that the shortwave direct solar component dominates the total radiative flux during daytime.

### 4.4.4 Calculation of plant canopy latent heat fluxes

An important part of the heat balance in the urban canopy represents the latent heat fluxes from the vegetation. The RTM explicitly computes the radiation balance for each grid cell of the volumetric plant canopy which allows to calculate the
evapotranspiration of this vegetation.

The evapotranspiration of the resolved vegetation is modelled using the Jarvis-Stewart method (Stewart, 1988) implemented following Daudet et al. (1999) on the leaf level. The leaf evapotranspiration depends on the leaf boundary layer conductance and the stomatal conductance. The leaf boundary layer conductance is a function of the wind speed (Daudet et al., 1999). The stomatal conductance is a function of the incoming shortwave radiation, the air temperature, the water pressure deficit and of
the relative soil water content following Stewart (1988)

After computing the evaporation per unit leaf area, the latent heat flux from leaves per the unit volume of vegetation is calculated by multiplication by the leaf area density $LAD$. The sensible heat flux is the residual of the energy balance, neglecting the storage.

### 4.4.5 Coupling to the radiation model

The radiative transfer model is coupled to the radiation model by providing effective radiation surface parameters to the radiation model, which are used as its boundary conditions.

The idea of these effective radiation parameters is that they would, when applied to a simple single surface, give similar radiation fluxes as the complex 3-D urban area. The three effective parameters are the effective surface temperature $T_{0,\text{eff}}$, the effective surface emissivity $\epsilon_{\text{eff}}$, and the effective surface albedo $\alpha_{\text{eff}}$.

To derive these effective parameters, the lower boundary conditions of the radiation model for both long- and shortwave radiation are considered as follows:

$$LW^{\uparrow} = \epsilon_{\text{eff}}\sigma T_{0,\text{eff}}^4 + (1 - \epsilon_{0,\text{eff}})LW^{\downarrow} \tag{90}$$







**Figure 2.** Instantaneous radiative fluxes at horizontal surfaces in an urban area (Prague-Dejvice), August 7, 2015 at 14:30 UTC (approx. 15:30 solar time). (a) diffuse longwave irradiance from the sky; (b) longwave irradiance from other surfaces (reflected and emitted); (c) total shortwave irradiance (direct and diffuse solar, reflected); (d) broadband net radiative flux (absorbed minus emitted).

$$SW^{\uparrow} = \alpha_{\mathrm{eff}} SW^{\downarrow} \,. \tag{91}$$





The energy conservation of longwave and shortwave radiation for the total urban area is used to derive $T_{\mathrm{eff}}$ and $\alpha_{\mathrm{eff}}$, while $\epsilon_{0,\mathrm{eff}}$ is chosen such that it represents the average of all urban surface emissivities. Details are given in Salim et al. (2019, to be submitted to this special issue).

### 4.5 Building surface model

The building surface model (BSM, formerly USM – Urban Surface Model, and including pavements) represents the counterpart of the LSM described above for building surfaces (i.e. walls and roofs). The core of the module represents calculation of surface energy balance together with propagation of the thermal energy inside the building material and energy exchanges with indoor and outdoor environments. Anthropogenic waste heat, e.g., from transportation or industry can be optionally prescribed by the user.

The initial version of the USM has been described in the paper Resler et al. (2017). The paper outlines the principles of the USM as well as it presents the first validation of the model against thermal observations from infrared camera for area of Prague–Holešovice. The current version of BSM model in PALM-4U 6.0 has been improved in several ways. The main improvement represents treatment of fractional surfaces. One fraction describes standard materials of the walls and roofs as in the original USM, the other fractions account for glass type of the surfaces (windows and other similar surfaces) and

green elements on facades and roofs. Other substantial improvement represents the treatment of humidity and latent heat flux similarly as in LSM (see Sect.3.5).

Most of the BSM calculations follow the methods outlined for the LSM in Sect. 3.5. The window tile takes the transmissivity of the glass material into account by calculating its optical depth and using the Beer-Lambert law. The individual heat transfer properties of the different window layers are neglected and equal layer properties are calculated using the overall thermal

transmittance. Green roofs (extensive or intensive) have underlying substrate layers where the temperature and heat transfer is taken into account. Green walls are considered facade-bound and are directly attached to the wall layers. The temperature and heat transfer within the material layers are calculated via the Fourier law of diffusion where the boundary condition on the outer surface is given by the surface energy balance while the temperature on inner surface can be either prescribed in the configuration or is calculated by the indoor and building energy demand model (see Sect.4.6).

The parameters of the urban canopy can be initialized with bulk parameters for specific building types or from a netCDF driver file (see Sect. 5.2.1). The current version of BSM also still includes the possibility to initialize these parameters through the legacy routines from CSV input files (see Resler et al., 2017). This option is, however, deprecated and will be removed in near future.

### 4.6 Indoor and building energy demand model

There is a strong interaction between the urban energy balance with the building energy balance. The urban atmosphere acts on the heat transfer through exterior walls, the longwave heat transfer, the solar heat gains and the ventilation. Considering also the internal heat gains and the heat capacity of the building structure, the energy balance of the interior building can be calculated based on an analytical solution of Fourier's law of heat conduction, see ISO 13790 (2008). The two main results are





the energy demand for heating and cooling and the indoor thermal environment. According to the building energy concept, the energy demand results in an (anthropogenic) waste heat, which is directly transferred to the air adjacent to the building. The indoor temperature is re-coupled via the building envelope to the urban environment and is affected indirectly by the urban atmosphere with a time shifted and damped temperature fluctuation.

Preliminary numerical and experimental studies clearly showed that different building concepts, their operation strategies and urban structures have a strong impact on the urban heat island effect. Furthermore, numerical studies revealed the reaction of the urban heat island effect on the building energy balance, see Voss and Künz (2012) and Pfafferott et al. (2011).

A holistic building model for the combined calculation of indoor climate and energy demand based on an analytic solution of Fourier's equation was implemented in PALM 6.0. The building model is integrated into the BSM (see Sect. 4.6). The interface
between these two models is the temperature in the building envelope (i.e., the interior wall, window, or roof temperature). The building energy supply system is simulated with simplified models for different heating, cooling, ventilation and/or air-conditioning concepts.

A commonly used database is used for the parameterization of both the facade and the building model, see Institut Wohnen und Umwelt (IWU). Furthermore, the building database provides building physical parameters of the building enve-
lope, geometry data and operational data. The building description is based on geometry, fabric, window and ventilation models for typical building types according to the year of construction or refurbishment, respectively. The user description is based on (stochastic) user behaviour regarding window opening and use of solar control, and user profiles regarding attendance, heat gains (i.e. plug loads and lighting), metabolic rate and clothing value. Energy efficiency factors and parameters related to the operation of the building energy system are defined for typical energy supply systems, e.g., district heating, boilers, heat pumps
or chillers.

The input information on building physical parameters from a regional survey or an urban planning tool is often uncertain and inconsistent. The model database is well-structured and includes sub-models which process information on different levels of accuracy and precision. Hence, the database is built up on a standardized building topology and can manually be adapted in order to evaluate measures with regard to the facade or to the building energy supply.

For details and evaluation of the the coupled indoor model, see Pfafferott et al. (2019, to be submitted to this special issue).

## 4.7   Surface spin-up mechanism

When using either LSM or BSM, or both, it is often difficult to initialize the material temperatures (i.e. temperature of the individual soil, pavement, and building wall layers) and the soil moisture for natural surfaces. This is primarily due to the fact that such input data is unknown (e.g. wall temperatures of buildings) or only sparse measurement data is available (soils and
pavements) from measurement campaigns. Furthermore, for idealized simulations, realistic initial material temperatures can have a strong effect on simulation results, particularly if only single diurnal cycles are simulated, which does not allow for a sufficient spin-up period of the material layers.

In order to overcome such issues, PALM offers a surface spin-up mechanism that allows for running long simulation periods as precursor to a full 3-D simulation. During such a spin-up period, the atmospheric code of the model is switched off and





only radiation model, LSM, BSM, and optionally the indoor model are activated. This reduces the computational costs during the spin-up phase by more than 90 %. The surface models are by design coupled to the atmosphere via the adjacent surface-parallel wind speed, temperature, and water vapor mixing ratio. During the spin-up period, it is thus assumed that all surface elements are exposed to the wind profile at model initialization, thus considering stationary synoptic conditions. Furthermore,

the wall-adjacent air temperature (i.e. $\theta_{\mathrm{mo}}$) is estimated by a simple sinusoidal diurnal cycle based on the cosine of the solar zenith angle that is calculated based on the geographical location, time, and two user-specific parameters:

$$\theta_{\mathrm{mo}}(t_{\mathrm{UTC}}) = \theta_{\mathrm{sp,av}} + \theta_{\mathrm{sp,amp}} \cdot \Psi(t_{\mathrm{UTC}} - 3600\,s)\,. \tag{92}$$

Here, $\theta_{\mathrm{sp,av}}$ and $\theta_{\mathrm{sp,amp}}$ are the mean temperature and its amplitude in the diurnal cycle, respectively. These must be set explicitly by the user. A time lag of of 1 h is imposed in order to account for a typical shift between maximum incoming solar

radiation and maximum temperatures in the diurnal cycle. Note that humidity in the atmosphere is currently considered to be constant in time during the spin-up phase and that there is no feedback of the surface scheme on the atmosphere during that period.

Figure 3 shows exemplary results from a spin-up for a test simulation with an urban setup and different surface types. The spinup simulation was 72 h and started on March 03. The surface forcing calculated based on Eq. (92) is shown in Fig. 3a

and reflects the time lag between maximum incoming radiation and maximum air temperature. Moreover, it is noticable that the maximum values of $SW_{\downarrow}$ increase each day, which reflects the calendrical progress. Figures 3b,c show the timeseries of the material temperatures below a surface element covered with grass and an asphalt pavement, respectively. For the soil temperature below a grass canopy, we note a maximum temperature amplitude at the first grid level in the soil (here 0.005 m) of about 10 K, while the amplitude within the respective asphalt pavement is much higher (about 22 K). Furthermore, we can

identify the expected time lag between different material layers, increasing with depth, and accompanied by a decrease in amplitude of diurnal temperature variations. Note also, that for the grass surface element, we note a stagnation or even drop in temperature in the afternoon hours. This is caused by a shadow cast by a nearby building and which is incorporated in the spin-up. For both materials, temperature fluctuations at a depth of 0.8 m are negligible within the simulated time period and significant diurnal variations are only present within the uppermost 0.2 m within the material. Moreover, we note that for both

surface types, the diurnal cycle of the uppermost material layer converges after 24 h of spin-up, which is a typical result based on our experiences. Furthermore, it is evident that the initial material temperature distribution in the uppermost layer deviates from the one after 72 h by 4 K and 6 K for short grass and pavement surfaces, respectively. This demonstrates the need of such a spin-up to achieve equilibrium conditions at model initialization. As an ad-hoc suggestion, especially for large simulation cases, where the spin-up becomes computationally expensive, we thus recommend to use at least 24 h of spin-up before starting

the 3-D model.

### 4.8 Self-nesting

For LES of urban ABL including the urban canopy, high grid resolution is needed. Xie and Castro (2006) have shown that at least 15–20 grid nodes along one typical building length are needed to satisfactorily resolve the most important turbulent

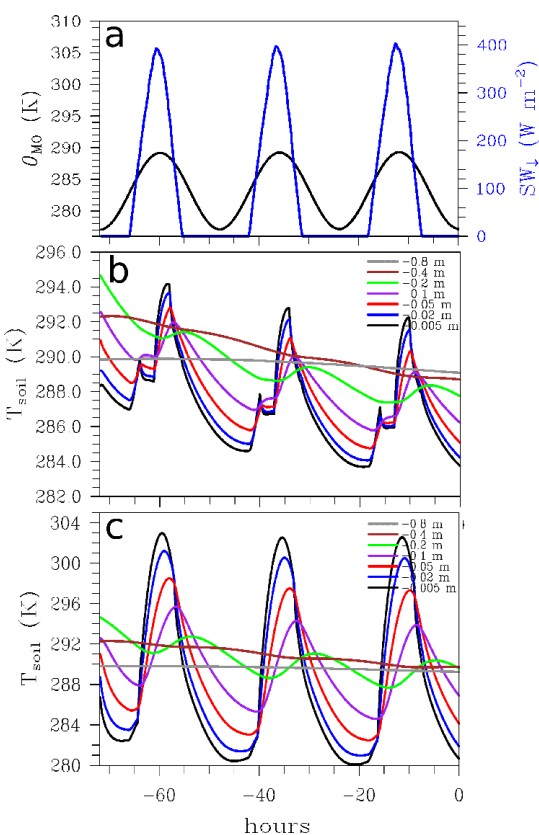

**Figure 3.** Timeseries during an exemplary surface spin-up of three days (72 h) starting on March 03 for a small LES setup, including different surface types and buildings, geographically located in Hannover, Germany. (a) shows the surface forcing, imposed by an unobstructed incoming solar radiation and the subsequent near-surface potential temperature forcing, while (b) and (c) show the resulting material temperatures $T_{soil}$ for a soil covered by short grass and an asphalt pavement, respectively. The time axis denote the hours until start of the full 3-D atmospheric simulation. Note that the radiative forcing shown in (a) does not match to the results shown in (b), where shadowing by buildings causes a drop in incoming shortwave radiation during afternoon hours.

structures within street canyons. To meet this requirement, the grid spacing should typically be in the order of 1 m, while at the same time the vertical extent of the model domain should cover the whole ABL and the horizontal extent should span over several ABL heights in order to properly capture the dominant turbulent eddies in the ABL. Moreover, the uncertainty related to the lateral boundary conditions usually decreases as the domain becomes larger. Therefore, even larger domains are

5 highly recommendable. However, using sufficiently high resolution in a sufficiently large domain often exceeds the available computational resources. In order to sufficiently resolve both, the entire ABL and also small-scale turbulent exchange processes within the areas of interest, a self-nesting capability was developed. We prefer self-nesting to adaptive grid techniques, because it fits to PALM's Cartesian grid structure, it is easier to optimize, and because the areas of interest where high resolution is required are always known in advance.





The idea of PALM's self-nesting is to simultaneously run a series of two or more LES model domains with different spatial extents and grid resolutions. The outermost model is called the root model. The other models are called child models and their domains are nested completely inside the root domain. The root model is a parent model and contains at least one child model. Child models can be recursively nested within each other, i.e., a child model can have its own child models for which it acts

as a parent model. A child model obtains boundary conditions for the prognostic quantities from its parent model through interpolation from the coarse to the fine grid. The self-nesting can be employed in a two-way or one-way mode. In the one-way coupled mode, only the child models obtain information from their parents while the flow in the parent model is not affected by the child model solution. In contrast, in the two-way coupled nesting (default), the parent model is influenced by its child models through so called anterpolation (Clark and Farley, 1984; Clark and Hall, 1991; Sullivan et al., 1996), where the fine-

resolution child solution is transferred back to the parent domain. The anterpolation is implemented using the post insertion approach (Clark and Hall, 1991), which means that the parent solution is replaced by the child solution restricted onto the parent grid in the domain of overlap.

The bottom boundaries of all model domains do always follow the terrain/building surface, hence the bottom boundary conditions are set in the usual way also for the child models. However, child model boundary conditions on the nested boundaries

(left, right, south, north and top boundaries) must be interpolated from the parent model. Concerning the interpolation method, it is important that the method conserves the mass-flow rate through the boundaries. If the mass conservation is violated in a two-way coupled run, a non-physical secondary circulation can develop. Clark and Farley (1984) developed a specific quadratic interpolation scheme that forms a reversible pair with the first-order anterpolation scheme they employed. This reversibility guarantees the mass conservation. In PALM, the interpolation algorithm has to cope with complex topography. Therefore we

did not select the quadratic scheme of Clark and Farley (1984). Instead, we use the simple and robust first order interpolation in which constant values are set to the child grid nodes residing within a parent grid cell. This scheme readily guarantees mass conservation. Another reason for using the first order interpolation is that it generally leads to smaller conservation error of momentum and scalar fluxes than higher-order schemes.

The anterpolation scheme is similar to that of Clark and Farley (1984). For any scalar variable the child grid values within

a parent grid cell are averaged and the averaged value is mapped to the parent grid node. For the momentum component the process is otherwise the same but because of the staggered-grid arrangement, the parent-grid cell is replaced by the two-dimensional cell face normal to the momentum component, i.e., the cell face where the momentum component is located. The anterpolation covers the whole volume occupied by the child domain except the parent-grid layers nearest to the nested boundaries.

The nested model system is implemented using two levels of Message Passing Interface (MPI) communicators. The inter-model communication is handled by a global communicator using the one-sided communication pattern (Remote Memory Access, RMA). The intra-model communication is two-sided and it is handled using a 2-D communicator. The intra-model communication system is the baseline parallelization of PALM (Maronga et al., 2015).

Beside an LES-LES self nesting, also a RANS-RANS nesting is implemented (one-way and two-way), where the parent and

the child models both solve the RANS equations and an interchange of $e$ as well as $\epsilon$ between parent and child is required.



Moreover, note that also a one-way RANS-LES nesting is currently under development, where the child model obtains RANS-filtered boundary conditions for its prognostic variables (except for $e$) from the parent model. This case, synthetic turbulence is imposed at the lateral child boundaries to trigger the development of turbulence in the child model.

An example of the self-nesting feature within an urban boundary layer is shown in Figure 4, where a child domain is nested

in an array of cube-shaped building blocks. The figure demonstrates that much more small-scale turbulence, which is mainly generated by the building blocks, can be resolved in the child domain.

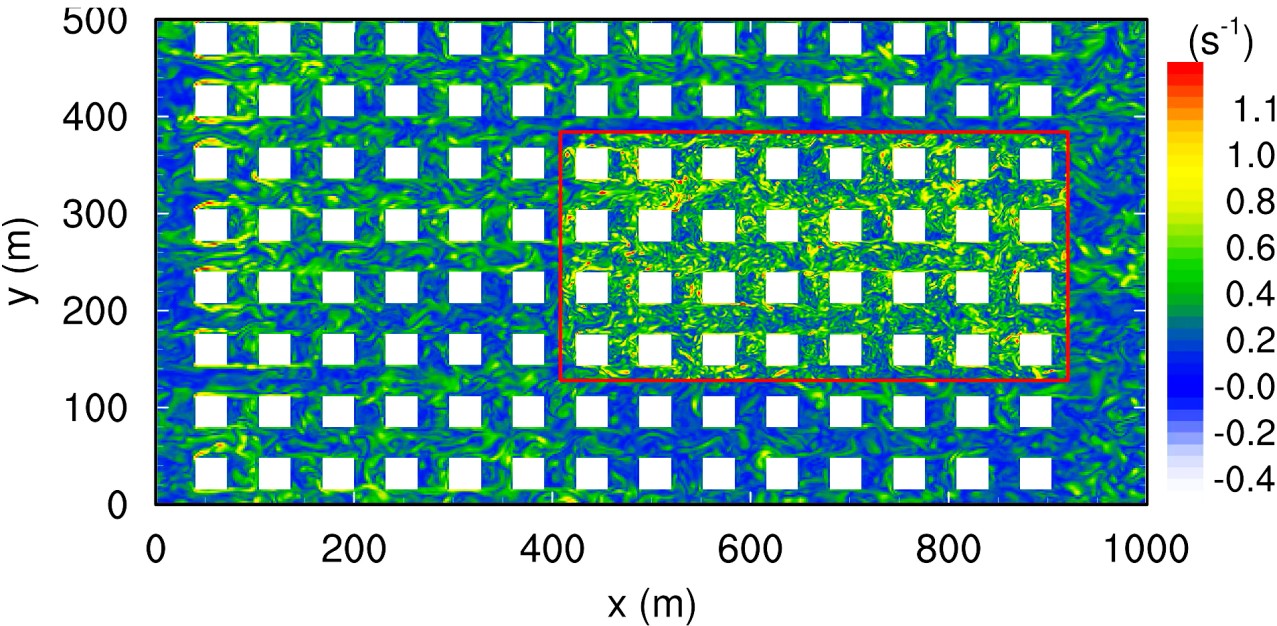

**Figure 4.** Instantaneous horizontal cross-section of the absolute value of the vorticity vector at a height of $10\,\mathrm{m}$ for a two-way LES-LES nested simulation of the turbulent flow around an array of cube-shaped buildings. The parent domain has a grid spacing of $2\,\mathrm{m}$ and the child domain (indicated by the red box) has a grid spacing of $1$ m. The mean flow is parallel to the $x$-axis from left to right.

For a detailed description of this self-nesting as well as sensitivity studies concerning, e.g., flow adjustment lengths within the child models, see Hellsten et al. (2019, to be submitted to this special issue). Furthermore, note that, while the 3-D self-nesting also can work in a 1-D manner, a separate method for pure 1-D nesting is implemented in PALM as described by

(Huq et al., 2018). In both cases, child and parent domains have identical horizontal dimensions and where the child obtains boundary conditions from the parent only at its top boundary. This 1-D self-nesting aims to improve the grid resolution near the surface throughout the entire horizontal extent of the domain.

### 4.9   Offline nesting

PALM has been successfully used to study non-stationary boundary-layer processes with both idealized and realistic setups. For

idealized setups with homogeneous flat terrain, cyclic boundary conditions are typically used at the lateral domain boundaries.



This approach allows to study boundary layers not only under quasi-stationary but also under evolving synoptic conditions, which can be represented by additional advective and nudging terms (Heinze et al., 2017). The usage of cyclic boundary conditions, however, becomes problematic for heterogeneous complex natural and urban terrain, where the flow characteristics may depend on the upwind surface conditions and wakes or circulations that are generated within the model domain may

re-enter the at the upstream boundaries and cause unrealistic flow feedbacks. Large buffer zones around the domain of interest are often required in such cases (Maronga and Raasch, 2013; Letzel et al., 2012).

In order to enable simulations of realistic heterogeneous domains under evolving synoptic conditions, PALM 6.0 offers non-stationary Dirichlet boundary conditions, which can be provided by mesoscale interface INIFOR. INIFOR is a stand-alone pre-processor that derives realistic initial- and boundary conditions for the PALM domain from the operational mesoscale

weather prediction model COSMO-DE/D2 (Baldauf et al., 2011). The pre-processed output is stored in a dynamic driver (see Sect. 5.2.1), a netCDF file that PALM reads continuously during the model run to apply non-stationary boundary conditions. INIFOR interpolates all required meteorological fields onto the PALM grid. This includes the three wind components $u_i$, water vapor mixing ratio $q_\mathrm{v}$, and the potential temperature $\theta$. The meteorological variables are provided as hourly boundary conditions at the four lateral and the model top boundary. Initial conditions are provided either in the form of vertical profiles

or as three-dimensional fields. INIFOR also derives the geostrophic wind profiles from COSMO's pressure field, which are used in the Boussinesq approximation to represent the evolving mesoscale pressure gradient, following the approach by Heinze et al. (2017). In addition to the meteorological variables, INIFOR also provides the initial soil temperature and moisture.

INIFOR can be used in combination with PALM's synthetic turbulence generator (see Sect. 2.3.3 to reduce the size of buffer zones that are needed to achieve fully-developed turbulence in the inner domain. The synthetic turbulence imposed at the

domain boundaries is continuously adjusted to the mean synoptic conditions. This is done by modifying the Reynolds-stress that is used as an input in the turbulence generator, which is computed following Rotach et al. (1996). This way, changes in the stability regime and the meteorological conditions that may alter the amplitude and length scales of turbulent fluctuations are reflected in the synthetic inflow turbulence.

With the time-dependent setting of lateral and top boundary conditions in PALM, an automated mesoscale offline nesting

of the PALM domain within the COSMO model is achieved. For a detailed description of INIFOR and the mesoscale offline nesting, a verification of this approach, as well as sensitivity studies regarding the derivation of geostrophic wind profiles, flow adjustment lengths, etc., see Kadasch and Suehring (2019) (to be submitted to this special issue).

### 4.10 Multi-agent system

Nowadays, more than half of the world's population lives in urban areas. Therefore, smart and sustainable city planning is more

and more required. In this process, also the comfort of pedestrians is regarded, which strongly depends on the surrounding atmospheric environment. Typically, human comfort and quality of live from a meteorological perspective are judged on the basis of 2-D-mapped indices (see Sect. 4.11). Taking these indices together with individual characteristics of a large number of pedestrians, like walking path and speed, age, clothing, etc., would enable the identification of areas for humans with high stress potential. Such hotspots cannot be determined from standard 2-D-maps because they do not take into account peoples'





behavior. Therefore, a Lagrangian based multi-agent system (MAS) was developed and integrated into PALM, based on the concept of the already existing Lagrangian particle model (see Maronga et al., 2015). Using this model, it is possible to simulate individual pedestrians (agents) moving as part of a crowd coupled to the atmospheric code with the aim to evaluate quality of life on an individual level and studying environmental effects on large groups of people. In addition, the model provides the

ability to simulate escape or evacuation scenarios coupled with the advanced method of LES used for turbulent dispersion of pollutants.

The MAS needs to perform a number of tasks to simulate the movement of pedestrians realistically. Firstly, agents must be able to navigate complex terrain to their individual targets, which includes the generation of a navigation graph from the Cartesian grid information present in PALM and a pathfinding algorithm for navigating the graph. Secondly, local interactions

of each agent with obstacles or other agents have to be considered.

The navigation graph is created in a preprocessing step from the Cartesian building configuration used in the PALM driver data. The tool calculating the graph (Agent Preprocessing Tool for PALM - APT-P) has been developed separately from PALM's model code. It is a standalone Fortran program, which is provided as a utility program shipped with PALM. The concept used in APT-P is called visibility graph. It is based on the idea that pedestrians use outer corners of obstacles as

navigation points for their movement. The agent will walk toward the next visible corner on their way to their final target, make a turn, and walk toward the next corner. Thus, the nodes of the visibility graph, indicating its physical locations, are the obstacle corners (see Figure 5). Obstacles are interpreted here as any static object that a pedestrian would try to avoid under normal circumstances, such as buildings, trees, or streets. Furthermore, the graph consists of connections between nodes if they are in line-of-sight of each other and the cost to travel between them. The procedure implemented in APT-P can be summarized

as follows: 1. The PALM building topography is converted to polygons (one per obstacle) containing all convex and concave corners as vertices. 2. The polygon data is simplified using the Douglas-Peucker algorithm (Hershberger and Snoeyink, 2006). 3. All remaining convex polygon vertices are added to the graph as nodes. 4. Connections are established between each pair of nodes that are in line-of-sight of each other, if the direct connection does not point into either of the involved obstacles. The associated travel cost is set to the Cartesian distance. And 5. The graph data is output to a Fortran binary file that can be read

by PALM.

During the simulation, the MAS uses the visibility graph for navigating each agent individually from its current position to its target. This is accomplished using a pathfinding method called A*-algorithm (Hart et al., 1968), which is able to find the optimal (fastest/shortest) path in a computationally efficient way concerning computation time and storage as well as to consider agent and target positions that were not previously stored on the visibility graph. To establish the path from the

agent's current position to its target, the following steps are taken: 1. The agent's current position and its target are added to the visibility graph. 2. The shortest path between these two points is calculated using A*. 3. The intermittent navigation points are shifted outward from the corresponding obstacle corner to a random position along a "gate" on the outer angle bisector of that corner to avoid collisions with obstacles or other agents. 4. Pathfinding (steps 1 through 4) is run again with each successive pair of navigation points along the path to avoid intersection of path sections with obstacles resulting from the outward shift.

Finally, 5. The resulting path is stored in the agent object as a series of intermittent targets.





For each agent, the direction towards its next intermittent target is calculated for each agent time step. The agent will accelerate toward those coordinates, and once close enough, the next intermittent target is calculated. At the final target the agent is deleted. Recalculation of an agent's path occurs when the agent has deviated to far from its current path, for example through repulsion by obstacles or other pedestrians. Figure 5 shows the final result after applying the pathfinding algorithm on
the visibility graph.

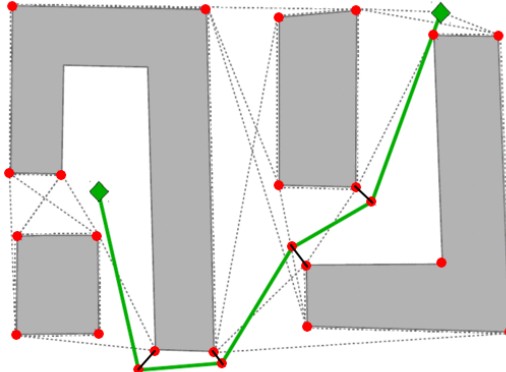

**Figure 5.** Final result of the pathfinding algorithm using the visibility graph created by the preprocessing tool APT-P. The dots at the obstacle corners mark the nodes of the graph and connections between these nodes, annotated with a cost to travel, are indicated by dashed lines. The left/right rhombus visualises the starting/end point of the path to go, which is illustrated by green lines that are connected at the shifted navigation points. The outward shift of the original nodes toward the actual navigation points is visualised by black lines.

Once the fastest path is found, the individual agent movement and close-range interaction with obstacles and other pedestrians is realized using a Lagrangian based social force model. Concepts from the original model formulation (Helbing and Molnár, 1995) as well as from its extension for close-range collision prediction and avoidance using a power-law approach (Karamouzas et al., 2014) have been adopted for the formulation of the MAS. The basic idea of the social force model is that
pedestrian movement is the result of all forces acting on the pedestrian caused by its surroundings and goals. These forces can be either repulsive (e.g., buildings, trees, or other pedestrians) or attractive (e.g., current target, shop windows, or shaded areas on a hot day). The resulting force on a pedestrian $\alpha$ determines its acceleration. For time integration, a forward Euler method is used. As human reactions take place on a very short time scale and due to the chosen social forces approach, the time step for the agent model must be very short (approximately 0.02 s - 0.04 s are recommended). In MAS, repulsion by obstacles and
other pedestrians as well as the acceleration force driving the agent toward its target are considered. The formulation of both repulsion forces is based on an exponentially decreasing repulsive potential of the building's wall and other pedestrians, respectively. However, a further repulsive force was added to simulate collision avoidance behavior based on a universal power law approach. It causes the agents to slow down, speed up or slightly alter their path to avoid colliding with each other. Finally, the acceleration force, which accelerates an agent toward their current target, is implemented using a relaxation time that describes





how quickly the pedestrian approaches the desired walking speed. For more information on the exact equations see Helbing and Molnár (1995) and Karamouzas et al. (2014).

### 4.11  Human biometeorology

The livability of cities might be defined through the well-being of the human population, which is affected and interacting with
the urban atmosphere. Biometeorological indices are a standard framework to assess this well-being (Staiger et al., 2019). The biometeorology module in PALM consists of two parts: A thermal comfort and a UV-exposure part. The thermal comfort part is allowing for the calculation of thermal indices approximating human thermal perception. Currently, the commonly used and well-validated indices Perceived Temperature (PT, Staiger et al., 2012), Universal Thermal Climate Index (UTCI, Jendritzky et al., 2012; Bröde et al., 2012) and the Physiologically Equivalent Temperature (PET, Höppe, 1999) are supported.

PT, UTCI and PET follow a similar approach of equivalent temperatures. They generally calculate the overall energy gain or loss caused by the prevailing meteorological environment and transfer this to an "indoor" reference environment with all parameters set to static pre-known values, except for the air temperature. The air temperature is then set to a value, that causes the same energy gain or loss, than the actual meteorological environment. The air temperature of the reference environment is the returned as the result for the index. While the general concept of the indices is similar, the actual implementation is
quite different. While, e.g., for PET the environments are compared through the energy balance, in PT the result of the simple indoor-only index Predicted Mean Vote (PMV, Fanger, 1972) are compared. UTCI is determined in a simplified way by a regression equation after Bröde et al. (2012). Further major differences can be found in the definition of the sample person, as well as in the consideration of clothing insulation. The three indices are provided for the one horizontal cell level, that is the closest possible to 1.1 m above ground level (the average human gravity center (Fanger, 1972)).

For the newly-developed MAS (see Sect.4.10), the thermal indices mentioned above cannot be used. They all do follow a steady-state approach, assuming long exposure of the sample person to constant environmental conditions (e.g. 2 hours for PT). Therefore a modified version of PT was implemented into the biometeorology module and coupled to the MAS. It aims at a better reproduction of the agents thermal stress caused by rapidly changing environmental conditions like radiation, wind speed, humidity and air temperature through consideration of the changes in the outer clothing surface temperature and a
storage term in the energy balance. Details of the thermal comfort model implementation are given by Fröhlich and Matzarakis (2019, to be submitted to this special issue).

With the exposure model of the UV-exposure part, calculations of biologically weighted human exposure in an urban environment can be performed. It is based on a three-dimensional voxel model of a human and the spectral radiance that takes into account the angular dependence of the radiation field (Seckmeyer et al., 2013). The spectral radiance was calculated by
the DISORT code of the UVSPEC model in the LibRadTran package (Mayer and Kylling, 2005). The model consists of the four main input parameters radiance, biological action spectrum (i.e., for erythema or vitamin D), human geometry and effects by obstructions. The human geometry is taken into account by using a 3-D model based on a computer tomography scan of an average human adult (Valentin, 2002). For the calculation of the UV exposure in different seasons, the human model can be clothed in various ways, e.g., with typical winter or typical summer clothing. In addition, the model considers the effect





of obstructions on the human exposure. Topographical elements (hills, vegetation, or buildings) cover various parts of the sky and hence block radiation from these directions (Schrempf et al., 2017). Therefore the effects by topographical elements are determined for each grid point. This enables the calculation of realistic maps of human exposure in an urban environment and shows for example the vitamin D weighted human exposure in dependence of the day time and season (i.e., position of the sun), viewing direction and the clothing of the human and the location within the city. An example of the exposure output is shown in Figure 6, where a map of the vitamin D production of a human in Berlin (Germany) is shown. Details of the exposure model implementation in PALM-4U are given by Schrempf et al. (2019, to be submitted to this special issue).

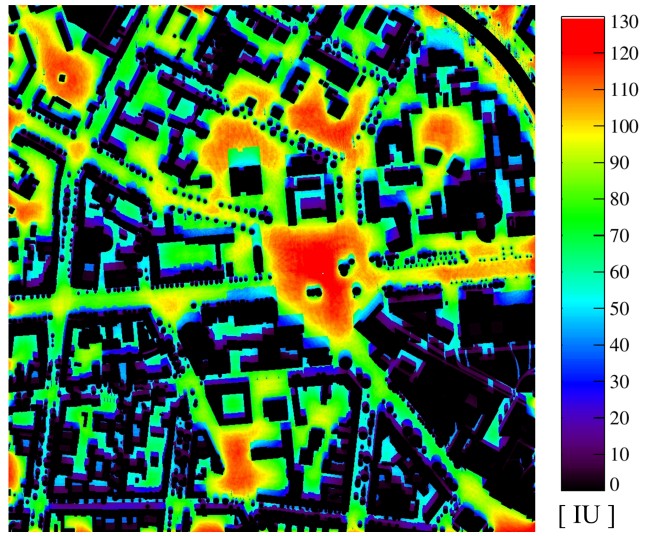

**Figure 6.** Vitamin D weighted exposure in IU per minute of a human on June 21 at noon in Berlin (Ernst-Reuter-Platz) under cloudless conditions. The human wears T-shirt and pants and is orientated towards south. Approximately 1000 IU per day are recommended as an adequate Vitamin D status.

## 5    Technical details

### 5.1    Data structure for surface elements

At wall-adjacent grid points additional code needs to be executed, e.g., for calculating wall functions or for the solution of the surface-energy balance. To efficiently access surfaces, we introduced a Fortran data structure that contains the relevant grid indices and the required surface variables, where all surface points are stored consecutively in one-dimensional arrays. In this way, additional surface-related code parts, e.g., for the LSM (see Sect.3.5) can be executed consecutively for all surfaces without adding 'IF-ELSE' statements within the main loops that run over all grid points of the 3-D grid, which would hamper loop vectorization and would reduce code legibility.





Surfaces can be horizontally aligned (facing upward or downward) or vertically aligned (facing northward, southward, eastward or westward). Beside its orientation, surfaces are further distinguished between default-type surfaces (i.e., non-interactive), natural-type surfaces (water-, vegetation-, pavement-covered, see Sect. 3.5), and urban-type surfaces (buildings, see Sect. 4.5). At default-type surfaces, no energy balance is solved and surface fluxes of sensible and latent heat can be either
directly prescribed or computed by MOST by prescribing $\theta_0$ and $q_0$. Note that the latter option is currently not implemented for downward and lateral-facing surfaces as here the stability correction via MOST has no physical foundation (stratification is always considered as in the vertical direction in MOST). For natural- and urban-type surfaces the respective surface fluxes are calculated by solving the energy balance using the embedded land- and urban-surface model (see Sects. 3.5 and 4.5, respectively). Surfaces with different orientation and type are treated individually, i.e., surface properties and all relevant information
are stored in individual data structures. Surfaces are automatically classified regarding their respective type and orientation depending on the input data (see Sect. 5.2.1), i.e., building and terrain height information.

### 5.2 Model operation and data handling

### 5.2.1 Model set-up via netCDF input data

The original topography model in PALM (see Sect. 4.1) was implemented around 2008, and was provided to the model via an
ASCII file containing the topography heights on a 2-D grid. The incorporation of full 3-D structures and interactive surfaces, however, requires setting of a great many of surface parameters like vegetation type, soil type, building height, building type, etc. for horizontal, but also for vertical walls. Moreover, the applications have increased significantly in terms of number of grid points, rending the ASCII input of topography height inappropriate and inconvenient. While ASCII input is still partly supported, a new netCDF interface was introduced in PALM 6.0, which allows to define all surface-related parameters that
do not change in time in a single netCDF file, the so-called static driver. This data can be provided for different LODs. For example, building heights can be prescribed as 2-D field as in the deprecated ASCII input (LOD 1), but it is also possible to prescribe fully 3-D topography via a 3-D byte field (LOD 2). Moreover, each vegetated surface element can be defined through setting one of 17 pre-defined vegetation types, which trigger automatic setting of the parameters required by the LSM, like roughness lengths, emissivity, and leaf area index (LOD 1). Additionally, some or all of these automatically set parameters can
individually be overwritten for each surface element (LODs 2 and 3).

In addition to static surface information, the stationary or time-depending meteorological forcing for PALM 6.0 can be provided in a separate netCDF file (so-called dynamic driver). The dynamic driver comprises all case-specific meteorological data, such as initialization data (e.g. wind and temperature as profiles or 3-D data, intial soil temperature and moisture, etc.), large-scale forcing tendencies or time-dependent lateral boundary data. The new pre-processing tool INIFOR (see Sect.4.9)
can be employed to generate such dynamic drivers automatically. The next model release will also contain a pre-processing tool to generate dynamic drivers for idealized scenarios.

For PALM's chemistry model, emission data can be provided in analogy to the dynamic driver in a separate file, the so-called chemistry driver, which contains information on chemical compounds and their emission distribution in space and time.



The PALM input data standard (PIDS) provides a technical documentation on the static, dynamic, and chemistry drivers [2].

### 5.2.2 Model steering

The PALM model system offers shell scripts to compile and run the PALM code, including preparation and submission of batch jobs, job chains, and the handling of I/O files. The old ksh-shell scripts `mbuild` and `mrun` have been replaced by

new bash-scripts `palmbuild` and `palmrun`. While the former scripts often required manual adjustments depending on the used MPI library and batch environment, the new scripts allow to run PALM on any batch system and with any MPI library without changing the scripts. All settings for the computing environment can now be provided via a configuration file `.palm.config.<ci>`, where `<ci>` is the so-called configuration identifier. The scripts can use configuration files for different computing environments (compilers, libraries, batch systems, etc.), which are selected by the script option `-c`, e.g.,

`palmrun -c intel_openmpi`. The organization of I/O files (file names, file types, folder structure, etc) is defined in a separate configuration file named `.palm.iofiles`. The complete script features and options are described in the online documentation.

### 5.2.3 Automatic model testing

The benefit of thorough software testing is higher software quality and reliability. In order to ensure high quality and reliability,

PALM is now equipped with a testsuite called `palmtest`. It is a python based software that automatically builds PALM according to specific build setups and executes PALM runs according to specific test cases. All the build setups and test cases are shipped within the PALM repository. The configuration of `palmtest` including all build setups and test cases is based on YAML files. Validation of the test results is done by comparing the generated results with reference files that are included in each of the test cases. These reference files are PALM output files and can be plain text files like the run control file as

well as netCDF files. `palmtest` is capable of performing restarts by declaring dependencies between different test cases and using the restart data of one test case to initialise another test case. Note that `palmtest` is not capable of dealing with a batch system as it is intended to be used locally on a developers computer to ensure a healthy commit. `palmtest` is also executed on a central server after each commit to ensure the overall health of the main repository.

### 5.2.4 Virtual measurements

Virtual measurements in flow simulations are often used to verify and validate model components by comparing model results with *in situ* measurement data (e.g. Maronga et al., 2014; Heinze et al., 2017; Resler et al., 2017). Vice versa, virtual measurements can help to uncover deficiencies in measurement strategies by imitation and evaluation of different strategies (e.g. Sühring and Raasch, 2013; Maronga, 2014; Sühring et al., 2019). The advantages of numerical simulations are, e.g., that boundary and background conditions are known, and by means of ensemble runs or idealized simulation setups, more realiable

statistics can be achieved. To emulate individual *in situ* observations at specific locations in realistically heterogeneous setups,

---

[2]Available as download from http://palm-model.org





however, a lot of manual work is required, e.g., to find out the respective grid indices. From the modeller's perspective, it is desirable to automatically translate observation coordinates into grid coordinates. And from the analysis perspective, measured and simulated data should have the same format, variable naming and unit conventions, etc., enabling usage of the same analysis tools for *in situ* and virtually-sampled datasets. For this purpose, PALM 6.0 provides a virtual-measurement module,

which requires standardized information from measurement campaigns via netCDF file (see Sect. 5.2.1), i.e., the geographical coordinates, the used reference system, the sampled quantities in the atmosphere, soil, or building walls, etc.. The input and output format is specified in the data standard defined within the [UC]$^2$ framework (see Sect. 5.2.6). The measurement coordinates for each site are translated into grid coordinates using the nearest model grid point. In order to minimize uncertainties due to biased geographical coordinates or errors in the model-surface or -topography setup, data is additionally sampled at the

surrounding grid points. Not only stationary measurement locations can be emulated, but also trajectory (mobile) observations such as drone, car, bicycle or pedestrian measurement systems. For a statistically more comprehensive picture, data is sampled along the entire trajectory at each model time step.

In order to avoid global communication during the simulation to merge the sample data for a site (for trajectory measurements the relevant grid points might be distributed over several cores), the sampled data is written to one individual binary file per

core. The PALM 6.0 model system provides a post-processor that merges the core-wise sampled data for each site and outputs the dataset into a separate netCDF file, resulting in one individual file for each measurement site.

### 5.2.5 Geographical information input data

In urban environments, PALM is designed to perform simulations from a very high spatial resolution of 1 m up to several tens of meters. The spatial resolution of the respective simulation determines the need of the spatial resolution of the geographical

information (GI) input data. Such data can originate from different data sources like satellite platforms such as Sentinel, aerial photographs, airbourne laserscanning data for high vegetation and buildings, and cadastral information. Additional, OpenStreetMap data can be used. The large heterogeneity and very different availability pose a challenge in the collection and processing of GI data to generate conform input data for PALM. Various techniques are required for preparation and conversion of geo data into the netCDF data format according to the PALM input data standard (see Sect. 5.2.1). This includes data fusion

techniques, a whole set of GIS methods and individual scripts and programs. Heldens et al. (2019, to be submitted to this special issue) provide details and a comprehensive insight how this has been achieved for selected cities in Germany. Some tools for data processing were already included in the PALM, such as a tree generator for converting information about single trees into 3-D leaf area density objects, and a python script for compiling static driver files based on rasterized GI data. Without doubt, the acquisition and preparation of suitable input data at the needed accuracy imposes one of the great challenges in

urban microscale modeling. In the future, more data preparation tools will thus be shipped with PALM 6.0 and automate these processes as far as possible.





### 5.2.6 Data output

Within the project framework of [UC]² a data standard was developed for measurement and simulation data sets which inherits most of the netCDF Climate and Forecast Metadata Conventions Version 1.7 (CF-1.7)[3] and is therefore also conform with the conventions of the Cooperative Ocean/Atmosphere Research Data Service (COARDS)[4] (Scherer et al., 2019b). The [UC]²

data standard describes precisely, among other parameters, the data type, naming of variables and meta data which must be included in the data sets. PALM 6.0 data output is conform to these data standards as all data output is available in netCDF format including all requested meta data. Naming of variables also comply to the data standard and is constantly adjusted as the data standard grows to includes further variables. In this context, PALM 6.0 data output now also can include geo-referencing (unlike prior PALM versions). Geo-referencing includes UTM coordinates as well as geographical longitude and latitude of

each horizontal grid point. Coordinates are calculated based on the given domain orientation and coordinates of a referencing point which is the front left grid point of the model domain.

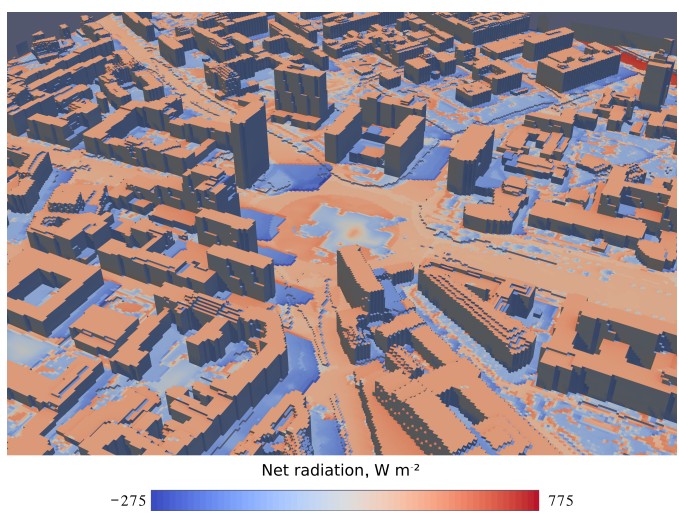

Net radiation, W m⁻²

−275                                                                                    775

**Figure 7.** Surface net radiation flux in a built-up city quarter (Ernst-Reuter-Platz, Berlin, Germany) on July 01 2009 at 14:00 UTC.

In a complex environment with buildings and/or hilly terrain, natural and artificial surfaces are unevenly distributed within the model domain, with each processor possibly treating a different number of surfaces with different face orientation within its subdomain. Surface data is output into netCDF files as one-dimensional arrays (see also Sect.5.1). To unambiguously identify

surfaces, also their grid coordinates as well as their orientation relative to zenith and azimuth is output. Further, to visualize surface data, e.g., the surface temperature or radiation fluxes, we additionally implemented an appropriate output format into PALM 6.0 based on the "Visualization Toolkit" (VTK) format (Schroeder et al., 2006). Therefore, as a first step, the vertices of the surface elements and the polygons spanned by these vertices, which unambiguously define the Cartesian wall/land-surface

[3]cfconventions.org/Data/cf-conventions/cf- conventions-1.7/cf-conventions.html

[4]see ferret.pmel.noaa.gov/Ferret/documentation/coards-netcdf-conventions





elements, are written to individual binary output files by each processor for its subdomain. Secondly, the surface data for an output quantity are gathered from the different surface types and orientations (see Sect. 5.1) at each output time step, and also written to individual binary output files by each processor. PALM 6.0 provides a post-processing tool to merge the binary output data from all subdomains and to convert these according to the VTK file format. This format enables to analyse and

visualize the surface data with well-known tools such as ParaView[5]. An example of the surface-data output is shown in Figure 7, where the surface net radiation flux is illustrated in a built-up city quarter in Berlin, Germany. Figure 7 indicates the shading of the direct solar radiation at horizontal surfaces and facades, as well as the effect of different surface properties on the surface radiation balance (see e.g. the region near the round-about in the center part of Figure 7).

### 5.3 Model optimization

Optimization efforts since PALM 4.0 were mainly targeted on allowing better code vectorization by compilers for Intel and AMD processor units. Within the red-black algorithm of the multigrid Poisson solver, the grid point values with even and odd indices are calculated alternately for each dimension of the 3-D-array. This results in loops programmed with a stride of two. On recent Intel processor generations, different flavors of vector units were implemented. Starting with AVX on Ivybridge, the current Skylake processor owns AVX512 units. Common to all AVX units is that scattered data access is either not implemented

or ineffective. In order to enable a vectorization of the red-black algorithm, a re-sorting of the $k$-dimension of the respective 3-D-arrays is done. For this, the values with even indices are stored in the lower half and the values with odd indices are stored in the upper half of the $k$-dimension vector. Now, loops can run with stride 1 along $k$ and the red-back algorithm vectorizes completely. This vectorization results in an overall speedup of more than 10% (for a test run with $256 \times 256 \times 128$ grid points in $x-$, $y-$, and $z-$direction, respectively).

### 6  Conclusions

In this overview paper, we gave an overview of the PALM model system 6.0. We described in detail the revisions made compared to PALM 4.0, which was described in the precursor paper of Maronga et al. (2015). As this paper was designed also as an overview paper for the PALM 6.0 special issue in this journal, we gave a rather brief summary of those parts for which companion papers will be submitted (or already have been) to this special issue. For all other new features and revisions, we

gave a more detailed description.

PALM 6.0 represents a tremendous enhanced and improved model system compared to its predecessor version. It does not only include improved model core physics like additional turbulence closure schemes, extended cloud microphysics, a wind turbine parameterization, and a fully interactive and coupled surface-radiation scheme, but also new components to enable in-depth simulations of urban environments (so-called PALM-4U components). Above all, this involves a building surface

and indoor model, gas phase chemistry and aerosol physics, and output of biometeorological indices. Furthermore, technical developments like two-way self-nesting and offline-coupling to the large-scale model COSMO enable large simulation domains

---

[5]https://www.paraview.org/





that allow to resolve the (turbulent) flow at very high spatial resolution in areas of special interest. While this is particularly useful for urban simulations, it will also be beneficial for other PALM applications.

While we focused on the technical description of the technical innovations in PALM 6.0, we did not address the topic of model validation in this paper in much detail, though it is of course a critical aspect in model development. During four

intensive observation periods in the framework of [UC]² (see Scherer et al., 2019a) an unprecedented database of measurement in urban area was created. Also, wind tunnel data was collected by the University of Hamburg, Germany. This database will be used to evaluate the new PALM-4U components based on large-scale PALM runs for the cities of Berlin, Hamburg, and Stuttgart. We will make use of the newly developed virtual measurement module to allow for one-to-one comparison with observational data. Further validation efforts are reported in the companion papers in this special issue.

Despite these model extensions, most of which developed in the context of the [UC]² programme, there is urgent need for further enhancements. For example, there is currently no parameterization for frozen water in PALM. Ice clouds as well as snow on surfaces or frozen water in the soil can thus not be simulated, imposing limitations for applications in regions prone to low temperatures and snow fall during winter time. Also, precipitation, though included for warm clouds in the PALM core, is currently not available together with urban surface configurations, above all, because of the missing incorporation in the RTM.

Moreover, several chemistry improvements regarding biogene volatile organic compounds, pollen transport, and improved aerosol description (including ultra fine particles) would be desirable. The multi-agent system might be further developed and it is planned to couple it to the traffic flow model MATSim (Horni et al., 2016) that allows for more detailed traffic emissions. Vehicle-induced turbulence (VIT) is known to be a key process for the dispersion of pollutants emitted by vehicles that is often parameterized in RANS-type models. For LES, however, there currently is no suitable parameterization available to account

for the additional mixing by vehicles. This could be either accounted for by explicitly placing moving car-like objects in the LES domain or by developing a new parameterization scheme suitable for LES models. Furthermore, important processes for urban climate studies are lacking, like a model to predict wind throw in stormy weather or sound emission and propagation, among others. Also, the Cartesian topography model in PALM currently implies that slanted roofs and walls are represented by step-like structures, which could be avoided by implementing an immersive boundary method (Mason and Sykes, 1978).

We aim to complete the model system by further developments as outlined above within a possible second funding phase of [UC]². In that course, we also plan to create a sustainable community model governance structure and will make significant effort to further strenghten PALM's position in the boundary-layer and urban climate scientific community.

*Code and data availability.* The PALM model system 6.0 is freely-available from http://palm-model.org and distributed under the GNU General Public License v3 (http://www.gnu.org/copyleft/gpl.html). For code management, versioning and revision control the PALM group

runs an Apache Subversion (http://subversion.apache.org) (svn) server. The PALM model system can be downloaded via the svn server, which is also integrated in a web-based project management and bug-tracking system using the software Trac (http://trac.edgewall.org). In this way, PALM users can use the web interface to browse through the code, view recent code modifications, and to submit bug reports via a ticketing system directly to the code developers. Furthermore, a model documentation, a detailed user manual as well as an online tutorial





are available on the Trac server and are constantly kept up to date by the PALM developers. Code updates and development is generally reserved to the PALM developers and supervised by the PALM administration at the Institute of Meteorology and Climatology at Leibniz University Hannover in order to keep the code structure clean, consistent, and uniform. However, we encourage researchers to contact us for collaborative code development that might be suitable to enter the default PALM model system. We also appreciate suggestions for future

PALM model system developments.

*Author contributions.*  The PALM model system 6.0 is a joint effort by several institutions and developers, most of which are authors of this paper. BM designed, prepared, and edited the manuscript with contributions by all listed coauthors.

*Competing interests.*  The authors declare that they have no conflict of interest.

*Acknowledgements.*  [UC]² is funded by the German Federal Ministry of Education and Research (BMBF) under grant 01LP1601 within the
framework of Research for Sustainable Development (FONA; www.fona.de), which is greatly acknowledged. The German Aerospace Center (DLR) Project Management supports the consortium. Benchmark and test runs with PALM have been performed at the supercomputers of The North-German Supercomputing Alliance (HLRN), which is gratefully acknowledged. The co-authors J. Resler, P. Krč, J. Geletič and V. Fuka were supported by the project URBI PRAGENSI CZ.07.1.02/0.0/0.0/16_040/0000383 under the programme OP PPR "Prague - Growth Pole of the Czech Republic" which is co-financed by EU. Some of the simulations were done on the supercomputer Salomon, which was
supported by The Ministry of Education, Youth and Sports of the Czech Republic from the Large Infrastructures for Research, Experimental Development and Innovations project „IT4Innovations National Supercomputing Center – LM2015070". The co-author Hellsten was supported by Academy of Finland (grant-agreement 277664) and the co-author Kurppa by the doctoral programme in Atmospheric Sciences (ATM-DP, University of Helsinki) and the Helsinki Metropolitan Region Urban Research Program.





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
