# Peer review of "Overview of the PALM model system 6.0"

_Geoscientific Model Development, 2019_

## Referee Comment (RC1) · Anonymous Referee #1 · 21 Jun 2019

Summary: This paper gives an overview of the PALM6.0 model, with a focus on the new advances made since PALM4.0. While I'm not an expert to judge every single aspect of the model development, my overall impressions are 1) the new modeling capabilities are very much needed by the scientific community and the authors should get all the credit for making a better model for the community, and 2) the paper is well written and has just enough details but not too much. I think the paper should be basically published as it is as a documentation and also an introductory reading. The only major suggestion I have is to create a table for all the acronyms as there are many of them. There are also a few minor issues from my reading, as follows:

1, I think dropping the overbar (or tilde which is used by the LES community) in some places but not others creates more confusion than the advantage it provides. I would recommend to follow the convention.

[Figure]

2, After equation 20, Pr can be adjusted. How?

3, The calculations of u*, u"w", and v"w". Are you sure they are internally consistent? If you use the horizontal wind velocity to compute u* with Eq. 28 as your line 25 (page 8) states, and use Eq. 32 to compute u"w" and v"w", do the results satisfy Eq.31? It does not appear to me this is the case.

4, line 15 page 27: have you defined BSM already? I couldn't find it before this line.

5, line 7 page 42: stratification is considered vertical does not explain why you don't do this for downward surface. It's also vertical.

6, line 15 page 45: is out should be are outputs

---

## Short Comment (SC1) · 14 Aug 2019

In my role as executive editor, this comment highlights an issue with the code availability in this manuscript which needs to be remedied before a revised manuscript can be accepted for publication in GMD.

Currently, the code availability section points to a project web site and a revision control repository (SVN in this case). Neither of these are permanent archive locations suitable for archiving the code associated with a journal publication. Please therefore archive the exact version of the software presented in this manuscript on a persistent, public archive. Most GMD users find Zenodo[1] a suitable choice, but other possibilities are available. Further information on arriving requirements is presented in the GMD
* * *
[1]https://zenodo.org

model code and data policy[2]. A more expansive description of the policy as well as the reasoning behind it is presented in the most recent GMD editorial[3].

[2]https://www.geoscientific-model-development.net/about/code_and_data_policy.html
[3]https://doi.org/10.5194/gmd-12-2215-2019

---

## Referee Comment (RC2) · Anonymous Referee #2 · 21 Oct 2019

In the manuscript "Overview of the PALM model system 6.0" the authors provide an exhaustive overview of the most recent version of the PALM model, an Open Source atmospheric microscale simulation model. The overview covers not only the components that are integral components of PALM, but also models that are distributed independently of PALM. PALM has been under development for a couple of decades and it represents a widely used, comprehensive, validated, and well documented microscale modeling capability. The manuscript focuses on model's capabilities without addressing their validation. The details of model components as well as their validation will supposedly be addressed in a number of separate manuscripts under preparation, or submitted to the special issue of Geoscientific Model Development Discussions.

General Remarks

An earlier version of PALM was already presented in the journal Geoscientific Model

Development, however, the new developments included in the version 6.0 are significant and warrant an update. Overall the manuscript is well written, but in an attempt to be exhaustive, it turned out excessively long. By referencing a number of manuscripts under review or in preparation for the special issue of Geoscientific Model Development Discussions the authors avoided addressing all the details, including model validation. However, without these references the manuscript would not stand on its own. These references include detailed description of modeling capabilities and should be published before the overview paper. Furthermore, the paper is already too long and it should be shortened by omitting description of standalone models (Sections 4.10 and possibly 4.11), also Subsection 5.2.5. has low information content and can therefore be omitted.

Taking all the above into account I do not recommend the manuscript for publication in the present form. The manuscript could be published in Geoscientific Model Development after suggestions for major and minor revisions are addressed and all the essential referenced manuscripts are accepted for publications.

Specific Remarks

Page 2, line 7 - Lilly 1967 is the first published paper about LES, however, it did not include any simulations, so it is not a proper reference here. Also, Deardorff published first LES results in 1970. Page 2, line 12 - Small scales are not parameterized, but the effect of small, unresolved scales on large, resolved scales. Page 3, line 14 - This sentence should start a new paragraph. Page 11, Equation (35) – Should it be $1/(\phi_M)^3$ or perhaps $\phi_H/(\phi_M)^3$? Page 12, line 6 - Basu and Lacser (2017) point to the potential problems with specifying surface fluxes and using the Monin-Obukhov similarity to determine surface temperature under stably stratified conditions. How is this addressed in PALM? Page 13, Table 4 – For integrated similarity function symbol $\psi$ is commonly used. Using capital $\Phi$ symbol for two different terms and differencing them by subscript can be confusing, it would be better to use a different symbol. Page 14, line 4 - It is not clear what is meant by "identical velocity and length

scales," perhaps for all the spatial directions - is that an assumption of isotropy? Page 15, line 4 - The sentence starting with "See Noh et al. (2004)..." is redundant it repeats the same information contained in the previous sentence. Page 15, line 16 – This should probably be "rain water mixing ratio" instead of just "rain water mixing." Page 19, line 30 – WRF-LES has already been used for some time with land surface models. Page 20, Table 5 – Perhaps a different symbol can be used for the source/sink terms. Page 22, line 22 – Papers that have not been published should not be cited. This comment applies to a number of references made related to yet to be reviewed and published manuscripts throughout this manuscript. Page 23, Equation (78) – The longitude, $\lambda$, needs to be expressed in radians not degrees, so it must be divided by 180 and multiplied by $\pi$. Page 25, Equation (84) – Is this equation correct? For $\psi = 0.5$ this results in a discontinuity in $\alpha_{sw,dir}$. Page 25, Section 3.5 "Wind turbine model" – The description of the ADM is imprecise and lacking. Page 25, line 24 – It would be more accurate to say that the actuator disk model accounts also for torque. Page 26, line 1 – The velocity, $U_{rel}$ is affected by local smearing of the drag, so it is not really representative of the free flow velocity assumed by an ADM. Page 31, line 8 – More information could be provided about the wall functions. Page 31, line 27 – A reference to a paper that has not been reviewed/published yet. Page 32, line 13 – A reference to a paper that has not been reviewed/published yet. Page 32, line 30 – A reference to a paper that has not been reviewed/published yet. Page 33, line 18 – A reference to a paper that has not been reviewed/published yet. Page 34, line 23 – Instead of "in pair" it should be "on par." Page 34, line 24 – It would be important to mention what the scaling is for RTM 1.0. Page 34, line 29 – It would be important to provide a reference. Page 36, Figure 2 – It would be better to use the same color scale for all four plots so that relative importance can visually obvious. Page 40, line 5 – Instead of "highly recommendable" better would be "strongly advised." Page 40, line 7 – It is not clear why is this called "self-nesting" and not just "nesting." Page 41, line 13 – Here, in "... domains do always..." "do" should be omitted. Page 41, line 31 – It is not clear what is meant by "model" here. Is it nest or physics model, or something entirely different? Page 42, line 8 – A reference to a paper that has not been reviewed/published yet. Page 43, line 27 – A reference to a paper that has not been reviewed/published yet. Page 43, Section 4.10 "Multi-agent system" - The paper is already very long and this may not belong in PALM description since the agent model is a standalone model. There are too many details about a standalone model, however, little is said about its utility and how it is coupled to PALM. Page 44, line 12 - Since it is a standalone program it is not clear that it should be included in description here. Perhaps it would be sufficient to mention it. Page 46, Section 4.11 "Human biometeorology" – Is this a standalone model? If yes perhaps it should be just mentioned and not described. Is this an online or an offline calculation? Page 46, line 25 – A reference to a paper that has not been reviewed/published yet. Page 47, line 7 – A reference to a paper that has not been reviewed/published yet. Page 47, line 13 – What about parallelization? Is this done for each part of the domain computation for which resides one specific processor core? In general, this could be written more clearly, perhaps with a few more details. Page 48, line 30 – Since pre-processing capability does not exist yet it is not clear that it should be mentioned. Page 50, line 8 – Is any interpolation used? Page 50, Subsection 5.2.5. – This section has low information content, and therefore could be omitted. Page 52, line 14 – Instead of "owns" it should be "includes." Page 52, line 20 – Section 6. "Conclusions" - Instead of "Conclusions" what is provided here is "Summary" and "Future Developments" or "Future Directions." It would be better to split this section in two sections. Page 53, line 54 – Instead of "immersive" more commonly used term is "immersed."

---

## Author Comment (AC1) · 12 Nov 2019

**Authors' responses to all reviewers**

First of all we would like to thank all reviewers for their comments and suggestions for improving the manuscript. We tried to take all of them into account in the revised manuscript. While referee #1 seems to be fine with the length of the manuscript, referee #2 finds the manuscript too long. We removed one section (5.2.5) as suggested by the referee and we also removed a few more paragraphs. This, however, did not lead to a significant shortening. Given the extent of the model described in this manuscript and the fact that this is an overview paper that shall show the full picture of the model, we believe the length is unavoidable. Also, we understand the paper as a technical look-up reference to help (potential) model users so that its length is not of major importance.

Please find our responses to all referee comments below. The changes made in the text can be also traced back in the attached marked-up manuscript. Please note that we also did some minor corrections based on comments provided by personal communication. In particular, these corrections are:

- homogenization of some text passages regarding references and use of abbreviations

- minor correction of some equations (e.g. in radiation model description)

- we removed all references to unpublished papers. In the introduction we state explicitly that "The individual new PALM-4U components, case studies, validation runs, and issues with suitable input data are presented and discussed in a series of companion papers in this special issue.", which appears to be sufficient for the purpose of this paper. In this way, we can push forward the publishing of the overview paper. This also allows for referencing the overview paper in the companion papers properly.

For details, see marked-up manuscript.

**Reply to Referee #1**

**Referee comment #1**

The only major suggestion I have is to create a table for all the acronyms as there are many of them.

**Authors' response**

We followed this suggestion and added a table for abbreviations used throughout the manuscript.

**Referee comment #2**

I think dropping the overbar (or tilde which is used by the LES community) in some places but not others creates more confusion than the advantage it provides. I would recommend to follow the convention.

**Authors' response**

We do not agree that a tilde is the standard for referring to filtered quantities (see e.g. the book of Wyngaard). However, we agree that omitting the overbar is confusion. We decided to add the overbar for all filtered quantities throughout the manuscript.

**Referee comment #3**

After equation 20, Pr can be adjusted. How?

**Authors' response**

What was meant is that the user can set a user-specific value instead of using the value of 1 which only holds for neutral stratification. We modified the text accordingly.

**Changes in the text**

p 9, l 9: "The Prandtl number can be changed to a user-specific value for different stability regimes."

**Referee comment #4**

The calculations of u*, u"w", and v"w". Are you sure they are internally consistent? If you use the horizontal wind velocity to compute u* with Eq. 28 as your line 25 (page 8) states, and use Eq. 32 to compute u"w" and v"w", do the results satisfy Eq.31? It does not appear to me this is the case.

**Authors' response**

Yes, the formulation is consistent using basic geometric considerations. Besides Eq. 28 and Eq. 31, we need the decomposition of $u_\mathrm{h}$ and $u_*$ into components, which is given by

$$u = \cos(\alpha)u_\mathrm{h} \ , \ v = \sin(\alpha)u_\mathrm{h} \tag{1}$$

where $\alpha$ is the angle between the $u-$component and the wind vector. The same decomposition is applied to $u_*$:

$$u_*^2\cos(\alpha) = \overline{u''w''} \ , \ u_*^2\sin(\alpha) = \overline{v''w''} \ . \tag{2}$$

Eq. 32 directly follows from calculating the derivative of $u$ (and $v$), and replacing using the above formulations together with Eq. 28. We do agree that this was not obvious in the manuscript. We hence added a sentence that the simple geometric decomposition is needed to end up with Eq. 32.

**Changes in the text**

p 10, l 24: "From Eqs. (28), (32), and a geometric decomposition of both the wind vector and $u_*$, it is possible to derive a formulation for the horizontal wind components, viz."

**Referee comment #5**

line 15 page 27: have you defined BSM already? I couldn't find it before this line

**Authors' response**

Corrected.

**Referee comment #6**

line 7 page 42: stratification is considered vertical does not explain why you don't do this for downward surface. It's also vertical

**Authors' response**

The reason is that in MOST gravitational acceleration is a stabilizing force for the turbulent exchange from an upward facing surface, while it is a accelerating force from a downward facing surface. The MOST relationships do not account for downward facing surfaces. For lateral surfaces, vertical gradients and gravitation are meaningless and thus cannot be used. We changed the text slightly to make this clear.

**Changes in the text**

p 47, l 1: "Note that the latter option is currently not implemented for downward and lateral-facing surfaces as here the stability correction via MOST has no physical foundation (stratification is always considered as in the vertical direction in MOST and gravitational acceleration is always acting as a restraining force)."

**Referee comment #7**

line 15 page 45: is out should be are outputs

**Authors' response**

Corrected.

**Reply to Referee #2**

**Referee comment #1**

An earlier version of PALM was already presented in the journal Geoscientific Model Development, however, the new developments included in the version 6.0 are significant and warrant an update. Overall the manuscript is well written, but in an attempt to be exhaustive, it turned out excessively long. By referencing a number of manuscripts under review or in preparation for the special issue of Geoscientific Model Development Discussions the authors avoided addressing all the details, including model validation. However, without these references the manuscript would not stand on its own. These references include detailed description of modeling capabilities and should be published before the overview paper. Furthermore, the paper is already too long and it should be shortened by omitting description of standalone models (Sections 4.10 and possibly 4.11), also Subsection 5.2.5. has low information content and can therefore be omitted.

**Authors' response**

See our replies to the specific comments below.

**Referee comment #2**

Page 2, line 7 - Lilly 1967 is the first published paper about LES, however, it did not include any simulations, so it is not a proper reference here. Also, Deardorff published first LES results in 1970.

**Authors' response**

We removed the reference to Lilly and added the reference to Deardorff (1970)-

**Referee comment #3**

Page 2, line 12 - Small scales are not parameterized, but the effect of small, unresolved scales on large, resolved scales.

**Authors' response**

The reviewer is right, we changed the text accordingly.

**Changes in the text**

p 2, l 13: "Turbulence scales larger than a chosen filter width are being directly resolved by LES models, while the effect of smaller turbulence scales on the resolved scales is fully parameterized within a so-called sub-grid scale (SGS) model."

**Referee comment #4**

Page 3, line 14 - This sentence should start a new paragraph.

**Authors' response**

Corrected.

**Referee comment #5**

Page 11, Equation (35) – Should it be $1/(\phi_M)^3$ or perhaps $\phi_H/(\phi_M)^3$?

**Authors' response**

The formulation is correct. $\phi_H$ drops out of the equation as the fluxes are prescribed and the vertical temperature gradient is not evaluated.

**Referee comment #6**

Page 12, line 6 - Basu and Lacser (2017) point to the potential problems with specifying surface fluxes and using the Monin- Obukhov similarity to determine surface temperature under stably stratified conditions. How is this addressed in PALM?

**Authors' response**

We are aware of the issue discussed by Basu & Lacser and we have recently developed an improved boundary condition to avoid issues in stable conditions (Maronga et al. 2019, BLM). This new formulation has not entered PALM yet. Besides, it is limited to applications over homogeneous surfaces. As PALM is most of the time used for applications involved some kind of heterogeneity (complex terrain, land surface heterogeneity, obstacles), this method cannot be used. While we agree that this is scientifically important issue, we feel that it does not fit into this model description paper. In order to account for the reviewer's comment, we added a short cautionary note instead.

**Changes in the text**

p 12, l9: "Also note that the above formulation can lead to violations of MOST for too coarse grid spacings in some cases, particularly for set-ups of stable boundary layers, as the first grid layer might be located in the roughness sub-layer of the surface layer. For a discussion of this issue and an improved boundary condition, see Basu and Lacser (2017) and Maronga et al. (2019b)."

**Referee comment #7**

Page 13, Table 4 – For integrated similarity function symbol $\psi$ is commonly used. Using capital $\Phi$ symbol for two different terms and differencing them by subscript can be confusing, it would be better to use a different symbol.

**Authors' response**

We agree. We replaced the symbols for integrated similarity functions with $\Psi$ and the additional terms using $\varphi$. In this way, all terms should be easy to differentiate.

**Referee comment #8**

Page 14, line 4 - It is not clear what is meant by "identical velocity and length scales", perhaps for all the spatial directions - is that an assumption of isotropy?

**Authors' response**

We agree that this sentence was not well-designed. We revised/corrected the text.

**Changes in the text**

p 14, l 4: "In order to apply the synthetic turbulence generator, information on the turbulent length scales for the three wind components in the x, y-, and z-direction, as well as the Reynolds stress tensor is required. These information can be either obtained from idealized precursor simulations or from observations (Xie and Castro, 2008). In combination with the offline nesting (see Sect. 4.9), PALM also offers the possibility to compute turbulent length scales and Reynolds stress following the parametrizations described by Rotach et al. (1996)."

**Referee comment #9**

Page 15, line 4 - The sentence starting with "See Noh et al. (2004) ... " is redundant it repeats the same information contained in the previous sentence

**Authors' response**

Respective sentence was removed.

**Referee comment #10**

Page 15, line 16 – This should probably be "rain water mixing ratio" instead of just "rain water mixing."

**Authors' response**

Corrected.

**Referee comment #11**

Page 19, line 30 – WRF-LES has already been used for some time with land surface models.

**Authors' response**

We are aware of this fact. However, we did not say that PALM is the first LES (it is not) having a dedicated land surface scheme. We believe our statement "traditionally" might be misleading and we changed the wording to "often".

**Referee comment #12**

Page 20, Table 5 – Perhaps a different symbol can be used for the source/sink terms.

**Authors' response**

$\Psi$ was replaced by $\chi$.

**Referee comment #13**

Page 22, line 22 – Papers that have not been published should not be cited. This comment applies to a number of references made related to yet to be reviewed and published manuscripts throughout this manuscript.

**Authors' response**

See general reply above.

**Referee comment #14**

Page 23, Equation (78) – The longitude, $\lambda$, needs to be expressed in radians not degrees, so it must be divided by 180 and multiplied by $\pi$

**Authors' response**

The reviewer is correct here. We corrected the equation. The same correction had to be made to latitude.

**Referee comment #15**

Page 25, Equation (84) – Is this equation correct? For $\psi = 0.5$ this results in a discontinuity in $\alpha\_sw, dir$.

**Authors' response**

There were parentheses missing in the equation. We corrected this.

**Referee comment #16**

Page 25, Section 3.5 "Wind turbine model" – The description of the ADM is imprecise and lacking.

**Authors' response**

A very detailed description of the ADM-R implemented in the PALM-WTM was not intended, given the broad scope of the manuscript. We tried, however, to enhance and improve the description – see the revised version of the manuscript.

**Referee comment #17**

Page 25,line 24 – It would be more accurate to say that the actuator disk model accounts also for torque.

**Authors' response**

We agree. The sentence was changed accordingly.

**Changes in the text**

p 25, l 22: "While in the frequently used simple version of the ADM (e.g. as proposed by Calaf et al. (2010)) the forces are uniformly distributed and only the thrust force is considered (thus ignoring the torque), the WTM provides an advanced ADM (ADM-R) that considers both thrust and torque as functions of the radial and tangential position on the rotor disk."

**Referee comment #18**

Page 26, line 1 – The velocity, $U\_rel$ is affected by local smearing of the drag, so it is not really representative of the free flow velocity assumed by an ADM.

**Authors' response**

$U_{rel}$ is the relative velocity a rotor blade would experience, so it is indeed affected by the smearing of the forces and even more by the induction of the rotor. It is not to be mistaken for the free flow velocity $U_{ref}$. While also a simple ADM has to derive the local velocity at the rotor by using an induction factor a ($U_{local} = (1-a)U_{ref}$), in an actuator line model (ALM) as well as in the ADM-R (which is based on the ALM) the induction by the rotor is implicitly accounted for as we directly use the local "measured" velocity. $U_{rel}$ is composed of the local flow components $U_x$ (axial component of the flow = velocity of the mean atmospheric flow considering the induction by the rotor) and $\Omega r - U_\theta$ (azimuthal velocity component including the motion of the rotor blades). We could derive $U_x$ from the free flow as in an ADM: $U_x = (1-a)U_{ref}$, but we rather used the actual local flow velocity which we think is more appropriate (even if slightly affected by the smearing) than the approximation via $U_{rel}$ (which is often difficult to determine) and $a$. This approach is also followed by other authors using a similar ADM-R, see e.g. Martinez-Tossas et al. (2015, Wind Energy, 18, 1047-1060).

**Referee comment #19**

Page 31, line 8 – More information could be provided about the wall functions.

**Authors' response**

The wording here was not perfect. What was meant is the calculation of the surface shear stress and other surface fluxes as it is described in Section 2.3. We revised the text accordingly.

**Changes in the text**

p 46, l 23: "At wall-adjacent grid points additional code needs to be executed, e.g., for calculating the surface shear stress (see Sect. 2.3) and for the solution of the surface-energy balance to determine surface fluxes of sensible and latent heat. To efficiently access "

**Referee comment #20**

Page 31, line 27 – A reference to a paper that has not been reviewed/published yet. Page 32, line 13 – A reference to a paper that has not been reviewed/published yet. Page 32, line 30 – A reference to a paper that has not been reviewed/published yet. Page 33, line 18 – A ref- erence to a paper that has not been reviewed/published yet.

**Authors' response**

See above.

**Referee comment #21**

Page 34, line 23 – Instead of "in pair" it should be "on par."

**Authors' response**

Corrected.

**Referee comment #22**

Page 34, line 24 – It would be important to mention what the scaling is for RTM 1.0

**Authors' response**

We changed the text accordingly

**Changes in the text**

p 34, l 18: "[...] and it represents a significant improvement over RTM 1.0, where the amount of view factors grew with $\mathcal{O}(n^4)$ and amount of canopy sink factors grew with $\mathcal{O}(n^5)$ in the worst case."

**Referee comment #23**

Page 34, line 29 – It would be important to provide a reference.

**Authors' response**

As the paper to be referenced has not been published yet, the only solution was to remove the respective sentence.

**Referee comment #24**

Page 36, Figure 2 – It would be better to use the same color scale for all four plots so that relative importance can visually obvious.

**Authors' response**

We actually tried this at first, but the problem is that it visually eliminates most of the spatial details. We even had to choose nonlinear scales with different coefficients to make all the details prominent. A multi-hue scale could provide more details over a large interval, but those are generally discouraged and explicitly prohibited in GMD as colorblind-unfriendly. We feel that it is perhaps more important to show the individual features in detail rather than to provide direct comparison between the absolute flux values of different components, which is not that significant spatially. We hence did not change the plots.

**Referee comment #25**

Page 40, line 5 – Instead of "highly recommendable" better would be "strongly advised."

**Authors' response**

Revised accordingly.

**Referee comment #26**

Page 40, line 7 – It is not clear why is this called "self-nesting" and not just "nesting."

**Authors' response**

Self-nesting refers to the model capability to nest an instance of the same model into the model, which is different from nesting a specific model into another model system. We added a sentence explaining what self-nesting means. Note also, that PALM has an offline nesting to large-scale models which should not be confused with the self-nesting.

**Changes in the text**

p 40, l 24: "Self-nesting here means that an instance of PALM can be nested into another instance of PALM."

**Referee comment #27**

Page 41, line 13 – Here, in ... domains do always ... "do" should be omitted.

**Authors' response**

Corrected.

**Referee comment #28**

Page 41, line 31 – It is not clear what is meant by "model" here. Is it nest or physics model, or something entirely different?

**Authors' response**

Model refers to an instance of PALM, let it be either a root model domain or a nested model domain. We added a sentence to ensure that this is clear to the reader.

**Changes in the text**

p 41, l 23: "The nested model system is implemented using two levels of Message Passing Interface (MPI) communicators. The inter-model communication (i.e. communication between different domains in a nested configuration) is handled by a global communicator using the one-sided communication pattern (Remote Memory Access, RMA). The intra-model communication, being is handled using the baseline parallelization of PALM, using a 2-D communicator and two-sided communication (Maronga et al. 2015)."

**Referee comment #29**

Page 42, line 8 – A reference to a paper that has not been reviewed/published yet. Page 43, line 27 – A reference to a paper that has not been reviewed/published yet.

**Authors' response**

See above.

**Referee comment #30**

Page 43, Section 4.10 "Multi-agent system" - The paper is already very long and this may not belong in PALM description since the agent model is a standalone model. There are too many details about a standalone model, however, little is said about its utility and how it is coupled to PALM.

**Authors' response**

The multi-agent system is a fully integrated part of PALM and no stand-alone model. We checked our text and believe that it does not suggest this in its current version. In order to have a comprehensive overview of PALM, it is required to provide a short overview in this paper. We thus decided to leave this part in the manuscript.

**Referee comment #31**

Page 44, line 12 - Since it is a standalone program it is not clear that it should be included in description here. Perhaps it would be sufficient to mention it.

**Authors' response**

We followed the suggestion of the reviewer and removed parts of the text.

**Referee comment #32**

Page 46, Section 4.11 "Human biometeorology" – Is this a standalone model? If yes perhaps it should be just mentioned and not described. Is this an online or an offline calculation?

**Authors' response**

All described components are integrated in PALM and no stand-alone programs. The calculate indicied online and output is generated directly by PALM. We adjusted the text slightly to avoid confusion. We see no need, however, to remove parts of the text.

**Changes in the text**

p 45, l 9: "[..] are supported, directly calculated in PALM, and output."

**Referee comment #33**

Page 46, line 25 – A reference to a paper that has not been reviewed/published yet. Page 47, line 7 – A reference to a paper that has not been reviewed/published yet.

**Authors' response**

See above.

**Referee comment #34**

Page 47, line 13 – What about parallelization? Is this done for each part of the domain computation for which resides one specific processor core? In general, this could be written more clearly, perhaps with a few more details.

**Authors' response**

We agree that the text was not precise enough. In fact, each processor core has its own array of surface elements that are independent from surface elements on other cores. Their interaction through radiative transfer is described and managed in the RTM. We revised the text to provide a more exact description.

**Changes in the text**

p 46, l 25: "At wall-adjacent grid points additional code needs to be executed, e.g., for calculating the surface shear stress (see Sect. 2.3) and for the solution of the surface-energy balance to determine surface fluxes of sensible and latent heat. To efficiently access surfaces, we introduced a Fortran data structure that contains the relevant grid indices and the required surface variables, where all surface points located on a specific subdomain (i.e. one processor core) are stored consecutively in

one-dimensional arrays. In this way, additional surface-related code parts, e.g., for the LSM (see Sect.3.5) can be executed consecutively for all surfaces without adding 'IF-ELSE' statements within the main loops that run over all grid points of the 3-D grid on the respective subdomain, which would hamper loop vectorization and reduce code legibility. On the Cartesian grid oriented to the cardinal directions, surfaces can be horizontally aligned (facing upward or downward) or vertically aligned (facing northward, southward, eastward or westward). Beside its orientation, surfaces are further distin- "

**Referee comment #35**

Page 48, line 30 – Since pre-processing capability does not exist yet it is not clear that it should be mentioned.

**Authors' response**

We removed this sentence and added a short sentence that INIFOR currently only supports COSMO data.

**Referee comment #36**

Page 50, line 8 – Is any interpolation used?

**Authors' response**

We are using the nearestneigbor method instead of inteprolating in space. We adjusted the text slightly to avoid confusion.

**Changes in the text**

ü 49, l 20: "The measurement coordinates for each site are translated into grid coordinates using the nearest model grid point instead of using interpolation to the exact site location. "

**Referee comment #37**

Page 50, Subsection 5.2.5. – This section has low information content, and therefore could be omitted.

**Authors' response**

We decided to remove the mentioned section.

**Referee comment #38**

Page 52, line 14 – Instead of "owns" it should be "includes. "

**Authors' response**

Corrected.

**Referee comment #39**

Page 52, line 20 – Section 6. "Conclusions" - Instead of "Conclusions" what is provided here is "Summary" and "Future Developments" or "Future Directions." It would be better to split this section in two sections.

**Authors' response**

We changed the title of the section to "Summary and future developments", but we think it is not necessary to divide the text into two separate parts.

**Referee comment #40**

Page 53, line 54 – Instead of "immersive" more commonly used term is "immersed."

**Authors' response**

Corrected.

**Reply to SC1**

**Short comment**

In my role as executive editor, this comment highlights an issue with the code availabil- ity in this manuscript which needs to be remedied before a revised manuscript can be accepted for publication in GMD. Currently, the code availability section points to a project web site and a revision con- trol repository (SVN in this case). Neither of these are permanent archive locations suitable for archiving the code associated with a journal publication. Please therefore archive the exact version of the software presented in this manuscript on a persistent, public archive. Most GMD users find Zenodo a suitable choice, but other possibilities are available. Further information on arriving requirements is presented in the GMD model code and data policy. A more expansive description of the policy as well as the reasoning behind it is presented in the most recent GMD editorial.

**Authors' response**

We have created a permanent archive using the data repository of the Research Data Repository or Leibniz University Hannover. The DOI of the model source code described in this paper is: 10.25835/0041607. We added a respective statement in the revised manuscript.